



# Decomposing reflectance spectra to track gross primary production in a subalpine evergreen forest

Rui Cheng[1], Troy S. Magney[1,11], Debsunder Dutta[2,12], David R. Bowling[3], Barry A. Logan[4], Sean P. Burns[5,6], Peter D. Blanken[5], Katja Grossmann[7,8], Sophia Lopez[4], Andrew D. Richardson[9], Jochen Stutz[8,10], and Christian Frankenberg[1,2]

[1]Division of Geological and Planetary Sciences, California Institute of Technology, Pasadena, CA, USA
[2]NASA Jet Propulsion Laboratory, California Institute of Technology, Pasadena, CA, USA
[3]School of Biological Sciences, University of Utah, Salt Lake City, UT, USA
[4]Department of Biology, Bowdoin College, Brunswick, ME, USA
[5]Department of Geography, University of Colorado, Boulder, CO, USA
[6]National Center for Atmospheric Research, Boulder, CO, USA
[7]Institute of Environmental Physics, University of Heidelberg, Germany
[8]Joint Institute for Regional Earth System Science and Engineering, University of California Los Angeles, Los Angeles, CA, USA
[9]Center for Ecosystem Science and Society, and School of Informatics, Computing, and Cyber Systems, Northern Arizona University, AZ, USA
[10]Department of Atmospheric and Oceanic Sciences, University of California Los Angeles, Los Angeles, CA, USA
[11]Department of Plant Sciences, University of California, Davis, CA, USA
[12]Department of Civil Engineering, Indian Institute of Science, Bangalore, India

**Correspondence:** Rui Cheng (rui.cheng@caltech.edu), Christian Frankenberg (cfranken@caltech.edu)

**Abstract.** Photosynthesis by terrestrial plants represents the majority of $CO_2$ uptake on Earth, yet it is difficult to measure directly from space. Estimating Gross Primary Production (GPP) from remote sensing indices is a primary source of uncertainty, in particular for observing seasonal variations in evergreen forests. Recent vegetation remote sensing techniques have highlighted spectral regions sensitive to dynamic changes in leaf/needle carotenoid composition, showing promise for tracking
ing seasonal changes in photosynthesis of evergreen forests. However, continuous daily measurements of spectrally resolved canopy reflectance are limited in these ecosystems. To investigate this potential, we continuously measured vegetation reflectance (400–900 nm) using a canopy spectrometer system, PhotoSpec, mounted on top of an eddy-covariance flux tower in a subalpine evergreen forest at Niwot Ridge, Colorado, USA. We analyzed driving spectral components in the measured canopy reflectance using both statistical and process-based approaches. The decomposed spectral components relate directly
to carotenoid pigments and co-vary seasonally with GPP, supporting the interpretation of the Photochemical Reflectance Index (PRI) and the Chlorophyll/Carotenoid Index (CCI). We show that using features from the entire 400-900 nm range show additional spectral changes near the red-egde but do not outperform the PRI or CCI indices for GPP predictions. In addition, we can quantitatively determine needle-scale chlorophyll to carotenoid ratios as well as anthocyanin contents using full spectrum inversions, both of which tightly correlated with seasonal GPP changes. Reconstructing GPP from vegetation reflectance using
Partial Least Squares Regression (PLSR) explained approximately 87% of the variability in observed GPP. Our results link





the seasonal variation of reflectance to the pool size of photoprotective pigments, highlighting all spectral locations within 400–900 nm associated with GPP seasonality in evergreen forests.

# 1 Introduction

Terrestrial Gross Primary Production (GPP), the gross $CO_2$ uptake through photosynthesis, is the largest uptake of atmospheric
$CO_2$ (Ciais et al., 2013), yet the uncertainties are large, hampering our ability to monitor and predict the response of the terrestrial biosphere to climate change (Ahlström et al., 2012). Hence, accurately mapping GPP globally is critical. In contrast to unevenly distributed ground-level measurements such as Fluxnet (Baldocchi et al., 2001), satellites can measure GPP globally and uniformly based on the optical response of vegetation to incoming sunlight. Remote sensing techniques have made progress on tracking photosynthesis using the absorption features of photosynthetic and photoprotective pigments (Rouse Jr et al., 1974;
Liu and Huete, 1995; Gamon et al., 1992, 2016). The progress is particularly important for evergreen forests, which have large seasonal dynamics in photosynthesis but low variability in canopy structure and color. However, these promising techniques still lack a comprehensive evaluation/validation using both continuous in-situ measurements as well as process-based simulations.

GPP can be expressed as a function of photosynthetically active radiation (PAR), the fraction of PAR absorbed by the canopy (fPAR) and Light-Use Efficiency (LUE):

$$GPP = PAR \cdot fPAR \cdot LUE, \tag{1}$$

with LUE representing the efficiency of plants to fix carbon using absorbed light (Monteith, 1972; Monteith and Moss, 1977). The accuracy of remote sensing derived GPP is limited by the estimation of LUE, which is more dynamic and difficult to measure remotely than PAR and fPAR, particularly in evergreen ecosystems. There have been several studies inferring the light absorbed by canopies (i.e. fPAR) from Vegetation Indices (VIs) that estimate the 'greenness' of canopies (Running et al.,
2004; Zhao et al., 2005; Robinson et al., 2018; Glenn et al., 2008), such as the Normalized Difference Vegetation Index (NDVI; Rouse Jr et al., 1974; Tucker, 1979), the Enhanced Vegetation Index (EVI; Liu and Huete, 1995; Huete et al., 1997) and the Near Infrared Vegetation Index (NIRv; Badgley et al., 2017). Current GPP data products derived from Eq. (1) rely on the modulation of abiotic conditions to estimate LUE (Xiao et al., 2004). LUE is derived empirically by defining a general timing of dormancy for all evergreen forests with the same plant functional type (e.g. Krinner et al., 2005) or the same meteorological
thresholds (e.g. Running et al., 2004). However, within the same climate region or plant functional type, forests are not identical - leading to uncertainties in estimated LUE (Stylinski et al., 2002; Gamon et al., 2016; Zuromski et al., 2018), which propagate to the estimation of GPP.

LUE in evergreen forests is regulated not only by abiotic conditions, but also biotic processes. Unlike deciduous forests, evergreen trees keep most of their needles throughout the entire year. Seasonal GPP variations in evergreens are thus driven
by LUE (Bowling et al., 2018). In order to regulate LUE, evergreen needles have to quench the excess absorbed sunlight through non-photochemical pathways - i.e. regulating the pool size of pigments involved in the dissipation of excess light, the xanthophyll cycle pigments (Adams and Demmig-Adams, 1994; Demmig-Adams and Adams, 1996; Zarter et al., 2006),





among other pigments (Verhoeven et al., 1996). While the regulation of excess energy dissipation is often done through several mechanisms, these pathways are summarized under the term non-photochemical quenching (NPQ).

Some vegetation indices are sensitive to photoprotective pigments (e.g. carotenoids) and can characterize the seasonality of evergreen LUE with some success. For instance, the Photochemical Reflectance Index (PRI; Gamon et al., 1992, 1997) and Chlorophyll/Carotenoid Index (CCI; Gamon et al., 2016) both use wavelength regions that represent carotenoid absorption features around 531 nm at the leaf level (Wong et al., 2019; Wong and Gamon, 2015a, b) and show great promise for estimating photosynthetic seasonality (Hall et al., 2008; Hilker et al., 2011a). Because of the stable canopy structure in evergreen forests,

CCI and PRI have been applied at the canopy level as well (Gamon et al., 2016; Garbulsky et al., 2011; Middleton et al., 2016). In addition, the green chromatic coordinate (GCC; Richardson et al., 2009, 2018; Sonnentag et al., 2012), an index derived from red-green-blue brightness levels of canopy images, is also capable of tracking the seasonality of evergreen GPP (Bowling et al., 2018). However, the full potential of spectrally resolved reflectance measurements to explore the photosynthetic phenology of evergreens has not been comprehensively explored at the canopy scale. The evaluation of pigment driven spectral changes in

evergreen forests over the course of a season is necessary to evaluate where, when, and why certain wavelength regions could advance our understanding of canopy photosynthetic and photoprotective pigments.

Here, we use continuous measurements in both spectral space (full spectrum between 400–900 nm) and time (sub-daily over an entire year) to evaluate the potential of hyperspectral canopy reflectance for better understanding VIs sensitive to pigment changes that regulate GPP in evergreen forests. Continuous measurements of spectrally resolved reflectance at the

canopy scale have so far been sparse at evergreen forest sites (Gamon et al., 2006; Hilker et al., 2011b; Porcar-Castell et al., 2015; Rautiainen et al., 2018). There are only a few empirical studies on hyperspectral canopy reflectance in evergreen forests (Smith et al., 2002; Singh et al., 2015). Yet, empirically decomposed canopy spectral reflectance has been used as a predictor of maximum photosynthetic capacity (Serbin et al., 2012; Barnes et al., 2017; Dechant et al., 2017; Silva-Perez et al., 2018; Meacham-Hensold et al., 2019), GPP (Matthes et al., 2015; Huemmrich et al., 2017; DuBois et al., 2018; Huemmrich et al.,

2019; Dechant et al., 2019), and other physiological properties (Ustin et al., 2004, 2009; Asner et al., 2011; Serbin et al., 2014).

In contrast to empirical methods, process-based approaches, such as canopy Radiative Transfer Models (RTMs) can help to quantitatively link canopy photosynthesis with leaf-level contents of photosynthetic/photoprotective pigments (Feret et al., 2008; Jacquemoud et al., 2009). With RTMs, we can use spectrally resolved reflectance to directly derive leaf pigment contents (Féret et al., 2017; Jacquemoud et al., 1995) and plant traits (Féret et al., 2019) .

Continuously observed reflectance makes it possible to differentiate NPQ pathways by comparing the methods mentioned above and VIs against far-red Solar-Induced Fluorescence (SIF), which is also available from our canopy spectrometer system, PhotoSpec (Grossmann et al., 2018). SIF is a very small amount of radiation, relative to reflectance, released via the de-excitation of absorbed photons by photosystem II (Genty et al., 1989; Krause and Weis, 1991). Steady state SIF is regulated by NPQ and photochemistry (Porcar-Castell et al., 2014) and provides complementary information on canopy GPP. Magney et al.

(2019) found the seasonality of photoprotective pigment content in a subalpine coniferous forest is significantly correlated with canopy SIF.





In the present study, we analyze continuous canopy reflectance data from PhotoSpec at a subalpine evergreen forest at the Niwot Ridge AmeriFlux site (US-NR1) in Colorado, US, and seek to understand the mechanisms controlling the seasonality of photosynthesis using continuous hyperspectral remote sensing. We first explore empirical techniques to study all seasonal

variations in reflectance spectra, identify specific spectral regions that best explain the seasonal changes in GPP, and then link these spectral features to pigment absorption features that impact both biochemical and biophysical traits. We also use full spectral inversions using a canopy RTM to infer quantitative estimates of leaf pigment pool sizes. Finally, we compare the spring onset captured by different methods and VIs to determine the underlying mechanisms of the methods/VIs that contribute to photosynthetic phenology.

## 2   Material and methods

### 2.1   Study site

The high-altitude (3050 m above sea level) subalpine evergreen forest near Niwot Ridge, Colorado, US, is an active AmeriFlux site (US-NR1, Lat: 40.0329 °N, Lon: 105.5464 °W; tower height: 26 m; Monson et al., 2002; Burns et al., 2015, 2016; Blanken et al., 2019). Three species dominate: subalpine fir (*Abies lasiocarpa var. bifolia*), Englemann spruce (*Picea engelmannii*), and

lodgepole pine (*Pinus contorta*) with an average height of 11.5 m, a leaf area index of 4.2 (Burns et al., 2016), and minimal understory. The annual mean precipitation and air temperature are 800 mm and 1.5 °C, respectively (Monson et al., 2002). The high elevation creates an environment with cold winters (with snow present more than half the year), while the relatively low latitude (40°N) allows for year-round high solar irradiation (Monson et al., 2002). Thus, trees have to dissipate a considerable amount of excess sunlight during the winter dormancy, which makes this forest an ideal site for studying seasonal variation of

NPQ including the sustained component of it during dormancy (Bowling et al., 2018; Magney et al., 2019).

### 2.2   Continuous tower-based measurements of canopy reflectance

PhotoSpec (Grossmann et al., 2018) is a 2D scanning telescope spectrometer unit originally designed to measure SIF. It also features a broad-band Flame-S spectrometer (Ocean Optics, Inc., Florida, USA), used to measure reflectance from 400 to 900 nm at a moderate (full-width-at-half-maximum = 1.2 nm) spectral resolution with a field of view (FOV) of 0.7° (more details

in Grossmann et al. (2018); Magney et al. (2019)). In the summer of 2017, we installed a PhotoSpec system on the top of the US-NR1 eddy-covariance tower, from where we can scan the canopy by changing both viewing azimuth angle and zenith angles. On every other summer day and every winter day, PhotoSpec scans the canopy by changing the view zenith angle with small increments at fixed view azimuth angles, i.e. elevation scans. Only one azimuth position is kept after Oct 18, 2017 to protect the mechanism from potentially damaging winter conditions at the site. Spectrally resolved reflectance was calculated

using direct solar irradiance measurements via a cosine diffuser mounted in the upward nadir direction (Grossmann et al., 2018) as well as reflected radiance from the canopy. The reflectance data used in this study are from Jun 16, 2017, to Jun 15, 2018.





Here, we integrated all elevation scans to daily-averaged reflectances (every other day before Oct 18, 2017) by using all scanning viewing directions with vegetation in the field of view over the course of a day, filtering for both low light conditions and thick clouds by requiring PAR to be both at least 100 $\mu$mol $m^{-2}s^{-1}$ and 60% of theoretical clear-sky PAR. A detailed description of data processing can be found in appendix B. To further test whether bi-directional reflectance effects impacted our daily averages, we compared the NDVI and NIRv at various canopy positions given a range of solar zenith and azimuth angles (appendix A). Neither of the VIs was substantially impacted by the solar geometry supporting the robustness of daily averaged canopy reflectance. About 49 winter days exhibited significantly higher reflectances, attributable to snow within the field of view, which we corroborated with canopy RGB imagery from the tower. After removing data strongly affected by snow and excluding the days of instrument outages, 211 valid sample days remained. The daily-averaged reflectance was computed as the median reflectance from all selected scans for a single day, which was then smoothed by a 10-point (3.7 nm) box-car filter over the spectral dimension (400 - 900 nm) to remove the noise in the spectra. Figure 1(a) shows the seasonally averaged and spectrally resolved canopy reflectances measured by PhotoSpec.

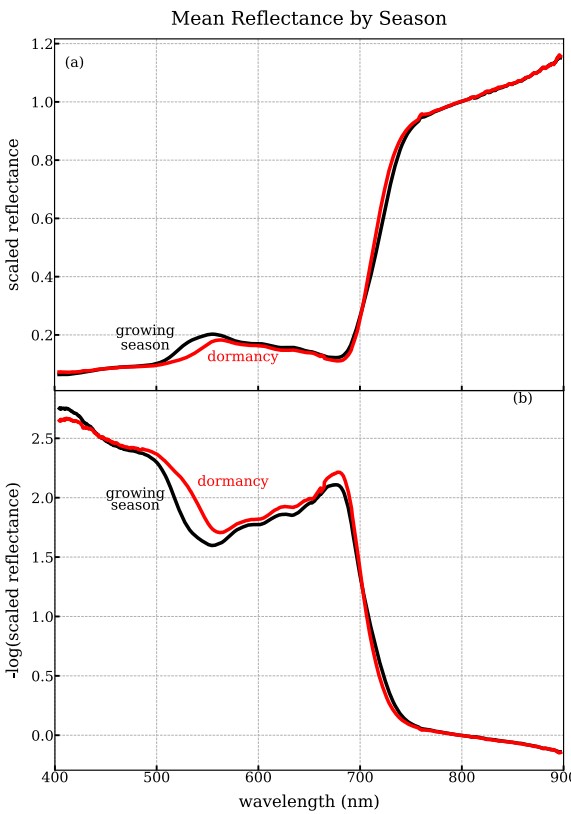

**Figure 1.** (a) Seasonally averaged canopy reflectance in winter dormancy (red) and the growing season (black) from PhotoSpec. (b) Seasonally averaged negative logarithm transformation of reflectance (400 - 900 nm). For comparison, we normalized the reflectance by the value at 800 nm on each day. Here, we refer to Nov 13 - Apr 18 as dormancy, and Jun 2 - Aug 21 ad the main growing season.





To further emphasize the change in reflectance as a result of changes in pigment contents, we transformed the reflectance (shown as R$_\lambda$) using the negative logarithm (Eq. (2)), as light intensity diminishes exponentially with pigment contents (Horler et al., 1983).

$$R_\lambda \propto exp(-C \cdot \sigma(\lambda)) \tag{2a}$$

$$C \cdot \sigma(\lambda) \propto -log(R_\lambda) \tag{2b}$$

with $\sigma$ = absorption cross section of pigments.

Therefore, the log-transformed reflectance (Fig. 1(b)) should correlate more linearly with pigment contents (shown as C). We also considered a variety of typical VIs using the reflectance data from PhotoSpec, such as:

$$NDVI = \frac{R_{800} - R_{670}}{R_{800} + R_{670}} \text{ (Rouse Jr et al., 1974)} \tag{3a}$$

$$NIRv = NDVI * R_{800} \text{ (Badgley et al., 2017)} \tag{3b}$$

$$PRI = \frac{R_{530} - R_{570}}{R_{530} + R_{570}} \text{ (Gamon et al., 1992)} \tag{3c}$$

$$CCI = \frac{R_{526-536} - R_{620-670}}{R_{526-536} + R_{620-670}} \text{ (Gamon et al., 2016)} \tag{3d}$$

$$GCC = \frac{R_{Green}}{R_{Red} + R_{Green} + R_{Blue}} \text{ (Richardson et al., 2009)}. \tag{3e}$$

In order to calculate GCC, we convolved the reflectance using the instrumental spectral response function (Fig. S1; Wingate et al., 2015) of the StarDot NetCam SC 5 MP IR (StarDot Technologies, Buena Park, CA, USA), which is the standard camera
model prescribed by the PhenoCam Network protocol (Sonnentag et al., 2012).

In addition to the reflectance measurments, we included relative SIF as well. Yang and van der Tol (2018) justified that the relative SIF, SIF normalized by the reflected radiation, is more representative for the physiological variations of SIF since it accounts for the complexity of signal due to canopy structure.

### 2.3 Eddy covariance measurements and LUE

An array of up and down-looking PAR sensors (SQ-500-SS; Apogee Instruments, Utah, US) above and below the canopy was used to calculate fPAR in half-hourly intervals. fPAR was smoothed with an 8-point (4 hour) running mean and 20-day running mean to remove the noise in the measurements. The first fPAR measurement started on Aug 8, 2017 (DOY 220).

Observations of Net Ecosystem Exchange (net flux of CO$_2$, NEE), PAR, and meteorological variables made at the US-NR1 tower are part of the official AmeriFlux Network data (Burns et al., 2016). GPP was estimated in half-hourly intervals
(Reichstein et al., 2005) using the REddyProc package (Wutzler et al., 2018), allowing us to compute LUE (Goulden et al., 1996; Gamon et al., 2016) at half-hourly intervals.

The light response curves show that GPP is a nonlinear function of PAR (Fig. 2; Harbinson, 2012). Magney et al. (2019) showed that fPAR does not significantly vary with seasons. At high light intensity, the carboxylation rate, driven by maximum





carboxylation rate ($V_{cmax}$), becomes the limiting factor (Farquhar et al., 1980). Thus, we defined the high-light/light-saturated

GPP (GPP$_{max}$) as the mean GPP at PAR levels between 1000 and 1500 $\mu mol\ m^{-2}s^{-1}$, a range which is covered throughout

the year, even in winter. Therefore, GPP$_{max}$ is less susceptible to short term changes in PAR. Yet, GPP$_{max}$ is proportional to

$V_{cmax}$ when the carboxylation rate limits photosynthesis. A higher GPP$_{max}$ thus indicates a greater $V_{cmax}$ and maximum

electron transport rate (J$_{max}$) when the variation of GPP$_{max}$ is independent from stomatal conductance and intercellular $CO_2$

concentration (Leuning, 1995). At low light intensity, photosynthesis is light limited. To implement Eq. (1), we defined the light

limited LUE (LUE$_{lightL}$) as the fitted slope of APAR against GPP at PAR between 100 and 500 $\mu mol\ m^{-2}s^{-1}$. We calculated

GPP$_{max}$ and LUE$_{lightL}$ from half-hourly GPP and APAR for each day. We also defined a more generalized effective daily LUE

(LUE$_{total}$) as the daily averaged ratio of GPP to APAR during the day. This effective daily LUE would be most applicable for

empirical LUE models that work on daily time-steps.

    We also included the meteorological variables provided from the AmeriFlux network data, such as Air Temperature (T$_{air}$)

and Vapor Pressure Deficit (VPD). Daytime daily mean T$_{air}$ and VPD were extracted from the half-hourly T$_{air}$ and VPD when

PAR was greater than 100 $\mu mol\ m^{-2}s^{-1}$.

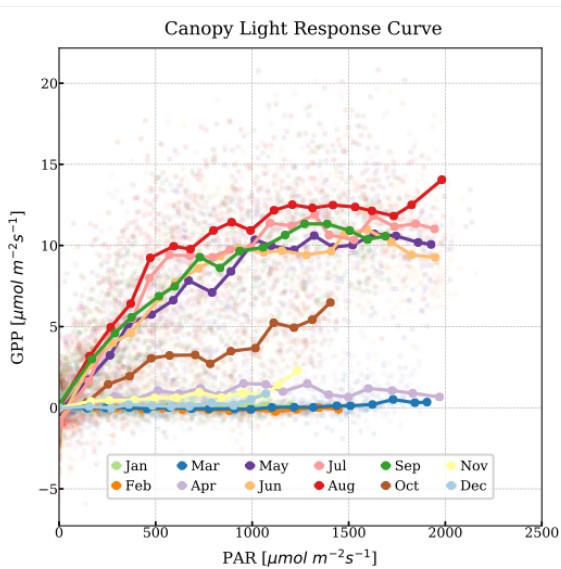

**Figure 2.** Half-hourly GPP as a function of PAR during the measurement period. Points are colored by month. Bold points are the median GPP when PAR is binned every 100 $\mu mol\ m^{-2}s^{-1}$ approximately. The solid lines represent the canopy light response curve.

## 2.4    Pigment measurements

To link canopy reflectance with variations in pigment contents, we used pigment data (Bowling et al., 2018; Bowling and

Logan, 2019; Magney et al., 2019) at monthly intervals over the course of the sampling period. Here, we focused on the total

xanthophyll content (Violaxanthin + Antheraxanthin + Zeaxanthin, V+A+Z) and total carotenoid content (*car*) measured on

*Pinus contorta* and *Picea engelmannii* needles with units of moles per unit fresh mass. *car* includes V+A+Z, lutein, neoxan-





thin, and beta-carotene. We also compute the ratio of chlorophyll to carotenoid contents (*chl:car*), because CCI derived from Moderate Resolution Imaging Spectroradiometer (MODIS) can track chl:car (Gamon et al., 2016). Overall, we can match 10 individual leaf-level sampling days for both pine and spruce samples with reflectance measured within ± 2 days.

### 2.5 Data-driven spectral decomposition

We assume that the spectrally resolved reflectance is a result of mixed absorption processes by different pigments. This allows us to apply an Independent Component Analysis (ICA; Hyvärinen and Oja, 2000) to decompose the log-transformed reflectance matrix (day of the year in rows and spectral dimension in columns) into its independent components. An advantage of the ICA is that it can separate a multivariate signal into additive subcomponents that are maximally independent, without the condition of orthogonality (Comon, 1994).The decomposed spectral components reveal characteristic features that explain most of the variance in the reflectance matrix. The corresponding temporal components show temporal variations of these spectral features. We extracted three independent components using the ICA algorithm (fastICA, python package scikit-learn v0.21.0), which explain more than 99.99% of the variance. The set of independent (spectral) components contain spectral shapes of pigment absorption features, which suggests that the corresponding temporal components are related to the variations of pigment contents. We will introduce the method of extracting pigment absorption features in a quantitative model-driven approach in section 2.6.

In addition to analyzing the transformed reflectance alone, we empirically correlated the reflectance with LUEs/GPP$_{max}$ using Partial Least Squares Regression (PLSR). PLSR is a predictive regression model which solves for a coefficient that can maximally explain the linear covariance of the predictor with multiple variables (Wold et al., 1984; Geladi and Kowalski, 1986). Applying the PLSR on the hyperspectral canopy reflectance and LUEs/GPP$_{max}$ results in a coefficient that emphasizes the key wavelength regions that contribute to the covariation. PLSR has been used to successfully predict photosynthetic properties using reflectance matrices in previous studies from the leaf to canopy scales (e.g. Serbin et al., 2012, 2015; Barnes et al., 2017; Silva-Perez et al., 2018; Woodgate et al., 2019). We used four PLSR components based on a four-fold cross-validation.

We implemented another set of PLSR analyses on the reflectance with individual pigment measurement as the target variable, such as the mean values of V+A+Z, car, and chl:car. We used a leave-one-out cross-validation because of the small sample size and found that two PLS components are optimal. Comparing the PLSR coefficient of pigment measurements at the leaf level with the PLSR coefficient of LUEs/GPP$_{max}$ connects the changes in LUEs/GPP$_{max}$ to the pool size of V+A+Z, because the reflectance is regulated by the absorption of pigments.

### 2.6 Process-based methods

PROSPECT+SAIL (PROSAIL, Jacquemoud et al., 2009) is a process-based 1-D canopy radiative transfer model (RTM) that models canopy reflectance, given canopy structure information (SAIL) as well as leaf pigment contents (PROSPECT) (Jacquemoud and Baret, 1990; Vilfan et al., 2018).



We use PROSAIL (with PROSPECT-D, Féret et al., 2017) to compute the derivative of the reflectance with respect to individual pigment contents, namely chlorophyll content (chlorophyll Jacobian, $\frac{\partial -log(R)}{\partial Cchl}$) and carotenoid content (carotenoid

Jacobian, $\frac{\partial -log(R)}{\partial Ccar}$). This helps explain the variations in reflectance decomposed from the empirical analysis.

    We also use PROSAIL to infer pigment contents by optimizing the agreement between PROSAIL-modeled reflectance and measured canopy reflectance from PhotoSpec. We fixed canopy structural parameters (e.g. the LAI to 4.2, as reported in Burns et al. (2015)) and fitted leaf pigment compositions as well as a low order polynomial for soil reflectance (appendix C), similar to Vilfan et al. (2018) and Féret et al. (2017). The cost function J in Eq. (4) represents a least-squares approach, where $\hat{R}$ is the

modeled reflectance.

$$J = \sum_{\lambda=450nm}^{800nm} (R_\lambda - \hat{R}_\lambda)^2 \qquad (4)$$

We used the spectral range between 450 and 800 nm, which encompasses most pigment absorption features.

## 3    Results and discussion

### 3.1    Seasonal cycle of GPP$_{max}$ and environmental conditions

The subalpine evergreen forest at Niwot Ridge exhibits strong seasonal variation in GPP, T$_{air}$, VPD, GPP$_{max}$, and PAR (which is still sufficiently high to drive photosynthesis in winter)(Fig. 3 and Fig. S2). Needles use light most efficiently at low light levels (LUE$_{light}$) for a fraction of the day. We start to observe a photosynthetic saturation at low PAR values ($\sim 500$ $\mu$mol $m^{-2}s^{-1}$; Fig. 2), which is represented by GPP$_{max}$, resulting in low efficiencies under high light conditions. LUE$_{total}$ represents the mean light use scheme throughout the day. Hence, LUE$_{light}$ was slightly higher than LUE$_{total}$ during most of the growing

season (Fig. S2). There were only a few days when PAR is so low that LUE did not reach light saturation for most of the day, when LUE$_{total}$ is more comparable to LUE$_{light}$. Therefore, we only show the results of GPP$_{max}$ in the rest of our analysis in the main text as it is more representative than LUE$_{lightL}$ and more physiology-driven than LUE$_{total}$. In addition, LUE$_{lightL}$ and LUE$_{total}$ have more missing data than GPP$_{max}$. Yet, due to the lower light intensity during storms, GPP$_{max}$ is not always available.

Abiotic factors play a strong role in regulating GPP$_{max}$ in this subalpine evergreen forest over the course of the season. For instance, as can be seen in Fig. S3, there is a strong dependence of GPP$_{max}$ with Tair, with an almost complete shutdown of photosynthesis during dormancy, even when the air temperature exceeds 5°C. During periods at the onset and cessation of photosynthesis, GPP$_{max}$ rapidly increases with temperature, potentially because needle temperature co-varies with T$_{air}$, and needle temperature controls the activity of photosynthetic enzymes which affects V$_{cmax}$. Spring warming approaches the

optimal temperature for photosynthetic enzymes, leading to activation of photosynthesis, while cooling in the early winter inhibits these enzymes (Rook, 1969). Once the temperature is around the optimum (in the growing season), T$_{air}$ is no longer the determining factor for photosynthesis. Melting snow with warming in spring made soil water available for uptake and caused the recovery of GPP$_{max}$ (Monson et al., 2005). When intercellular $CO_2$ concentration is not a limiting factor, GPP$_{max}$ is more representative of Vcmax and does not vary with PAR and T significantly.





**Figure 3.** Time series of Air Temperature ($T_{air}$), Vapor Pressure Deficit (VPD), Photosynthetically Active Radiation (PAR), Gross Primary Production (GPP), and $GPP_{max}$. DOY 166 (2017) is the first day of observation. The vertical dashed line divides the observations from Day of Year (DOY) for year 2017 and 2018.





**Figure 4.** A set of three spectral components (top) and corresponding temporal components (bottom) from ICA decomposition. The first two spectral components are overlaid with the chlorophyll Jacobian ($\frac{\partial - log(R)}{\partial Cchl}$, dash-dotted) and the carotenoid Jacobian ($\frac{\partial - log(R)}{\partial Ccar}$, dotted). The third spectral component is overlaid on the annual mean shape of transformed reflectance spectra. Temporal components are overlaid with GPP$_{max}$ (grey line). The axis of Jacobians is not shown because its magnitude is arbitrary here. The vertical dashed line divides the observations from DOY for year 2017 and 2018.

## 3.2 Seasonal cycle of reflectance

In Fig. 4 the Jacobians show the maximum sensitivity of the reflectance spectral shape to carotenoid content at 524 nm, and near 566 nm and 700 nm for chlorophyll. The first peak of the chlorophyll Jacobian covers a wide spectral range in the visible, while the second peak around the red edge is narrower. It can be seen that the first spectral ICA component has a similar shape as the chlorophyll Jacobian. The corresponding temporal component has a range between -0.2 to 0.2 without any obvious seasonal



variation, consistent with a negligible seasonal cycle in chlorophyll content as shown in the pigment analysis. However, there
is a gradual increase before DOY 50 in the first temporal component, which appears to be anti-correlated with the temporal
component of the second ICA structure. Two major features in the second spectral component can be observed. One is a
negative peak centered around 530 nm, which aligns with the carotenoid Jacobian. At the negative logarithm scale, the negative
values resulting from the negative ICA spectral peak multiplied by the positive ICA temporal components indicate there are

fewer carotenoids during the growing season. Conversely, positive values resulting from a negative spectral peak multiplied
by the negative temporal components indicate there are more carotenoids during dormancy (i.e. sustained photoprotection
via the xanthophyll pigments; Bowling et al., 2018). Another feature has a valley-trough shape, which is co-located with the
chlorophyll Jacobian center at the longer wavelength at the red-edge. The center of this feature occurs at the shorter-wavelength
edge of the chlorophyll Jacobian but does not easily explain changes in total chlorophyll content, which should show equal

changes around 600 nm. The corresponding temporal component apparently varies seasonally with $GPP_{max}$. In addition, the
second temporal component transitions more gradually from dormancy to the peak growing season than $GPP_{max}$. Unfortunately,
we are missing data to evaluate the relative timing of $GPP_{max}$ cessation. The third spectral component is similar to the mean
shape of reflectance spectra. Its temporal component holds around zero throughout the year. Overall, the second ICA spectral
component is more representative of the seasonal variation in the magnitude of total canopy reflectance than the other spectral

components. The spectral changes around the red-edge in the second component is interesting and might be related to structural
needle changes in chlorophyll-a and chlorophyll-b contributions (de Tomás Marín et al., 2016; Rautiainen et al., 2018), which
are not separated in PROSPECT.

  CCI, PRI, and GCC (Fig. 5(a-c)) follow the seasonal cycle of $GPP_{max}$ closely. CCI and PRI use reflectance near the center
of the 530 nm valley feature, the spectral range that is most sensitive to the change of carotenoid content, so that they match

changes in $GPP_{max}$ very well. PRI is the smoothest throughout the year, without any significant fluctuations within the growing
season, as is observed in $GPP_{max}$, which co-varied with $T_{air}$ and VPD. This performance is excellent given that PRI was
originally developed to track short term variation in LUE (Gamon et al., 1992). The GCC also correlates well with $GPP_{max}$, but
less than CCI and PRI. As can be seen in Fig. S1, the peak of the green channel used for GCC is close to the carotenoid Jacobian
peak, while the red channel feature covers a part of chlorophyll Jacobian feature. This explains the sensitivity of the GCC to

changes in both carotenoid content as well as chlorophyll. The bands used in GCC uses are broader than the ones used by
PRI and CCI, however it still captures these variations and can be computed using RGB imagery. Gentine and Alemohammad
(2018) found that the green band helps to reconstruct variations in SIF using reflectances from MODIS. While they speculated
that most variations in SIF are thus related to variations in $PAR \cdot fPAR$ (Gentine and Alemohammad, 2018), we show here
that the green band indeed captures variations in LUE as well. NDVI (Fig. 5(e)) and NIRv (Fig. 5(f)) do not have an obvious

seasonal variability. Similar to the ICA components, all VIs are quite noisy during dormancy. This noise may be due to snow
because we only removed the reflectance when the canopy was snow covered. Scattered photons possibly still reached the
telescope when there was snow at the ground, which is true for our study site as snowpack commonly exists in winter (Bowling
et al., 2018).





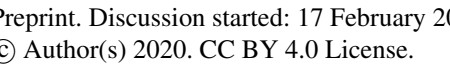

**Figure 5.** Magenta points are time series of VIs: (a) CCI, (b) PRI, (c) GCC, (d) relative SIF (e) NDVI (f) NIRv.The grey points in the background show GPP$_{max}$. The Pearson-r$^2$ values of regressing VIs and GPP$_{max}$ are noted in each plot. The p values of all correlations in this figure are less than 0.005. The vertical dashed line divides the observations from DOY for year 2017 and 2018.



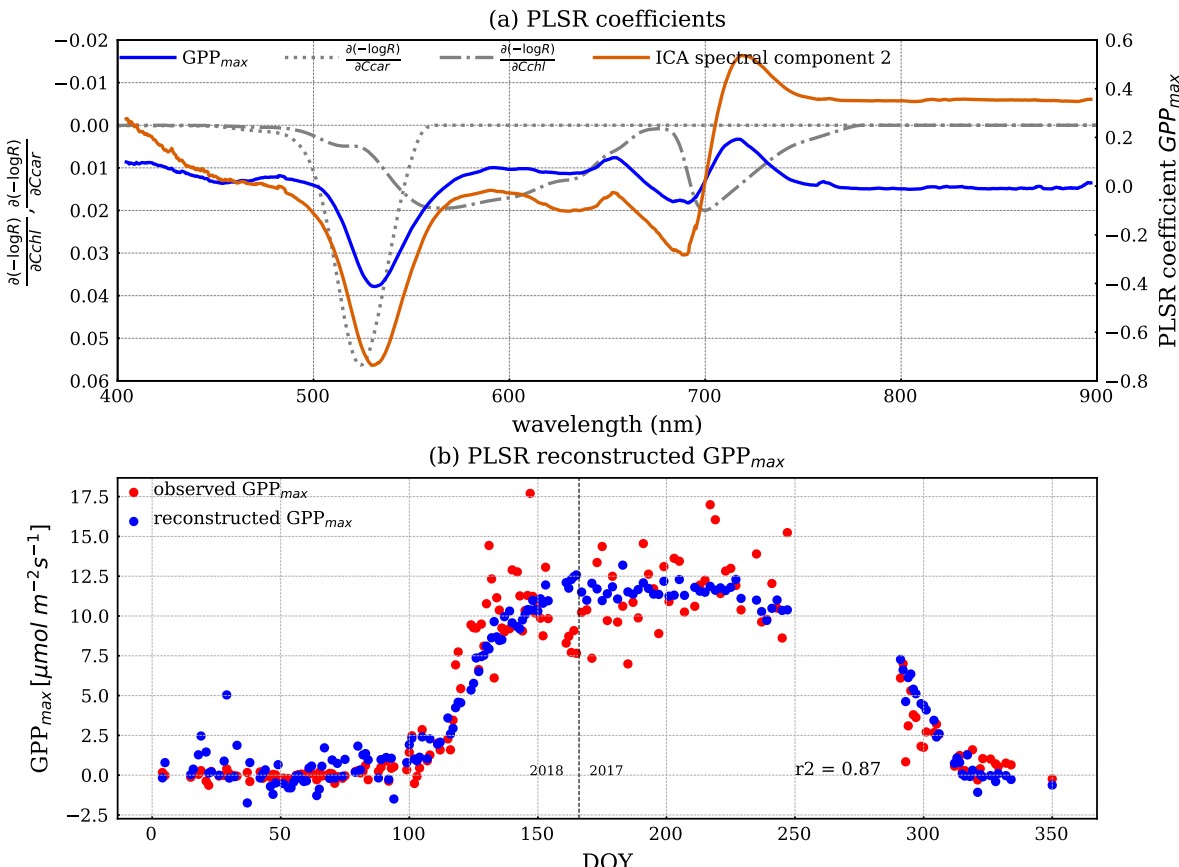

**Figure 6.** (a) PLSR coefficients of reflectance with GPP$_{max}$. The overlaid dash-dotted and dotted lines are chlorophyll and carotenoid Jacobians, respectively. The overlaid solid line is the second ICA spectral component. (b) The reconstructed GPP$_{max}$ (blue) by PLSR is overlaid with the observed GPP$_{max}$ (red). The vertical dashed line divides the observations from DOY for year 2017 and 2018.

### 3.3 PLSR coefficients of reflectance with GPP$_{max}$ and pigment measurements

The spectral shape of the PLSR coefficient with GPP$_{max}$ highlights a peak (centering at 532.0nm) near that of the carotenoid Jacobian with the same valley-trough feature observed near the second peak of the chlorophyll Jacobian (Fig. 6(a)). All the PLSR coefficients are similar (Fig. S4) because LUEs/GPP$_{max}$ are similar in terms of the seasonal trend.

    The reconstructed GPP$_{max}$ captures the transition period, while the noise in reflectance during dormancy propagates to the reconstructed GPP$_{max}$. The high-frequency variations in GPP$_{max}$ during the growing season are not captured by any of the
methods (Fig. 6(b)), which indicates those high-frequency variations are not related to pigment content, but rather changes in environmental conditions that lead to rapid changes in photosynthesis. Overall, the observed GPP$_{max}$ is significantly correlated with the PLSR reconstruction (Pearson-$r^2$=0.87), but very similar compared to CCI and PRI. A similar PLSR model of re-



flectance but with pigment measurements (Fig. 7) shows a direct link between pigment contents and reflectance. It can be seen that the PLSR coefficients of reflectance are very similar, irrespective of the target variable. They feature a valley near the peak

of the carotenoid Jacobian and a valley-trough feature near the peak at the longer wavelength of chlorophyll Jacobian. This spectral shape is also very similar to the second ICA spectral component and PLSR coefficients of $GPP_{max}$. V+A+Z, chl:car, and car are all nicely reconstructed by using the PLSR coefficients and reflectance (Fig. 7(b)). The reconstructed V+A+Z, car, and chl:car are correlated with the measured ones with Pearson-$r^2$ values of 0.84, 0.71 and 0.93, respectively.

    The second ICA component and PLSR empirically show the seasonality of reflectance using two different empirical frame-

works. ICA only used the reflectance, while the PLSR model accounts for variations in both reflectance and $GPP_{max}$ or pigment content. Yet, both ICA and PLSR agreed on similar spectral features that covary seasonally with LUEs/$GPP_{max}$. This indicates that the resulting spectral features are primarily responsible for reflecting this seasonal cycle. The overlap of these features with the chlorophyll/carotenoid absorption features shows that the seasonality of $GPP_{max}$ is related to variation in pigment content at the canopy scale, which is directly validated with similar PLSR coefficient of reflectance and pigment contents. These results

are consistent with leaf-level measurements of chlorophyll contents, which were shown to not change throughout the year; meanwhile, a higher ratio of chlorophyll to carotenoid content during the growing season was detected in this forest (Fig. 7).

### 3.4   Process-based estimation of pigment content

PROSAIL inversion results further supports the link between canopy reflectance, pigment contents and $GPP_{max}$. Figure 8 shows a continuous time-series of $Cchl$, $Ccar$, Anthocyanin content ($Cant$), and $\frac{Cchl}{Ccar}$ as derived from the PROSAIL canopy RTM

inversion model. Examples of simulated and measured reflectance spectra shown are in Fig. C1. Anthocyanins are another type of photoprotective pigment (Pietrini et al., 2002; Lee and Gould, 2002; Gould, 2004) that protects the plants from high light intensity (Hughes, 2011). The pigment inversions closely match the seasonality of $GPP_{max}$. $\frac{Cchl}{Ccar}$ shows the greatest sensitivity in capturing the seasonal cycle, with the strongest correlation to leaf level measurements. The inversed $Cchl$ has the weakest empirical relationship with the measured one (Fig. 8(a) right panel). Apparently, some of the inversion errors of individual

$Ccar$ and $Cchl$ contents cancel out in the ratio, making the ratio more stable. $Cant$ performs similarly as $Ccar$, since they both are photoprotective, and the anthocyanins absorb at 550 nm (Sims and Gamon, 2002), which is close to the center of carotenoid absorption feature. Even though we lack field measurements of anthocyanins to validate anthocyanins retrievals, the inversions show that more than just carotenoid content can be obtained from full-spectral inversions.

    Strictly speaking, the complex canopy structure of evergreens makes the application of 1D canopy RTMs such as PROSAIL

difficult (Jacquemoud et al., 2009; Zarco-Tejada et al., 2019). Yet, Moorthy et al. (2008); Ali et al. (2016); Zarco-Tejada et al. (2019) reasonably discussed the pigment retrieval in conifer forests with careful applications. In our study, the reflectance was collected from needles with a very small FOV, and our study site has a very stable canopy structure throughout a year (Burns et al., 2016). Thus, the inversion results are meaningful for discussing the seasonality of pigment contents. In the future, radiative transfer models that properly describe conifer forests, such as LIBERTY (Dawson et al., 1998), could be used.





Figure 7. (a) PLSR coefficients of reflectance and three pigment measurements. The overlaid dash-dotted and dotted lines are chlorophyll and carotenoid Jacobians, respectively. The overlaid solid line is the second ICA spectral component. (b) The reconstructed pigment measurements (blue) by PLSR is overlaid with the measured mean pigment measurements (red). The error bar is one standard deviation of the measurements. The vertical dashed line divides the observations from DOY for year 2017 and 2018.

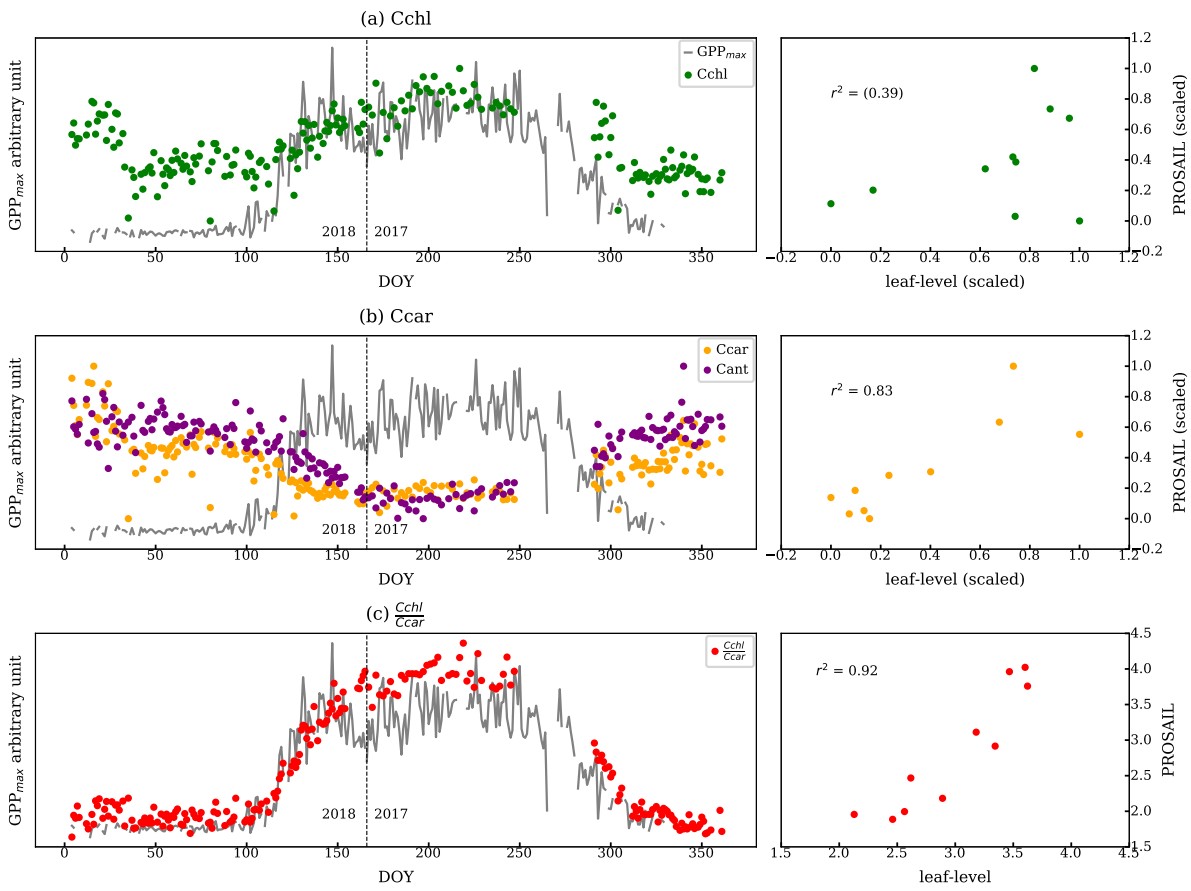

**Figure 8.** The left panels are the estimations of (a) *Cchl*, (b) *Ccar*, *Cant* and (c) $\frac{Cchl}{Ccar}$ from the PROSAIL overlaid with the GPP_max. We normalized two metrics because they report the pigment contents in different units. The vertical dashed line divides the observations from DOY for year 2017 and 2018. The plots on the right compare the pigment contents using PROSAIL and leaf-level measurements. The correlations are statistically significant except *Cchl*.





## 3.5 Comparison across methods

Although decomposing the hyperspectral canopy reflectance and using relative SIF (Fig. 5(d)) both successfully track the seasonal cycle of evergreen LUE, they underlie different de-excitation processes. During the growing season, environmental conditions primarily drive the high-frequency variations in LUEs/GPP$_{max}$. Relative SIF responds to such environmental stresses (van der Tol et al., 2014) so that it appears to track seasonal and diurnal variations better than reflectance.

There is also some variability between reflectance-based methods and relative SIF during the transition periods between the growing season and dormancy. We focus on the growing season onset since the reflectance measurements are not available during the cessation period. The onset (DOY 60 to 166) described by all the methods mentioned above as well as the relative SIF from Magney et al. (2019) are compared in Fig. 9, using a sigmoid fit to available data. The observed GPP$_{max}$ has the most rapid yet latest growing onset. The methods and VIs derived from or related to the pigment contents increased earlier than GPP$_{max}$ - such as the ICA component, PLSR coefficient, PROSAIL $\frac{Cchl}{Ccar}$ and CCI. However, they build up slowly to reach the maximum, which suggests that reduction of the carotenoid content is a slower process than the recovery of LUE. Reflectance-based VIs (Fig. 5) and decomposing methods (Fig. 4 and 8(b,c)) have a slower growing season onset than GPP$_{max}$, as found in Bowling et al. (2018) as well. On the other hand, relative SIF started the onset at almost the same time as the GPP$_{max}$, and it quickly reached the maximum. Therefore, using both SIF and reflectance to constrain the LUE prediction (van der Tol et al., 2014) can further improve the prediction accuracy.

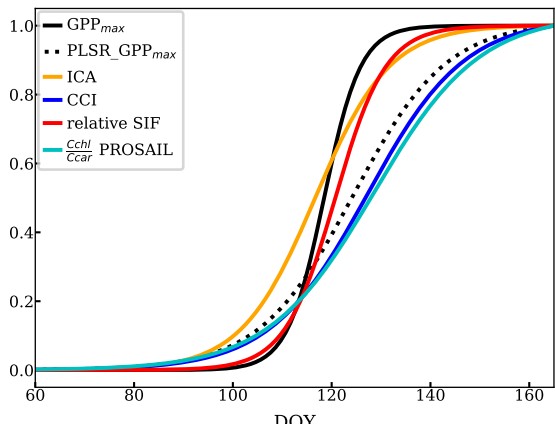

**Figure 9.** Temporal evolution of of the growing season onset using sigmoid fits (scaled) of PLSR, ICA, CCI, chlorophyll to carotenoid ratio and relative SIF.



## 4   Conclusion and future work

In this study, we analyzed seasonal co-variation of GPP and the spectrally resolved VIS-NIR reflectance signal, as well as several commonly used VIs. The main spectral feature centered around 530 nm is most important for explaining the seasonal cycle of reflectance (400 – 900 nm) and LUE, which corresponds to changes in carotenoid content. This explains why CCI, PRI,
and GCC track GPP seasonality so well, as most variations are driven by carotenoid pool changes. Our analysis included RTM simulation and in-situ pigment measurements throughout the season confirms the link between reflectance/VIs and pigment contents. The comparison of reflectance/VIs and relative SIF reveals differences in the timing of the growing season onset, pigment changes and SIF, indicating the potential of using both reflectance and SIF to track the seasonality of photosynthesis. However, the close correspondence between both SIF and reflectance suggest that hyperspectral reflectance alone provides
mechanistic evidence for a robust approach to track photosynthetic phenology of evergreen systems. Because seasonal variation in pigment concentration plays a strong role in regulating the seasonality of photosynthesis in evergreen systems, our work will help to inform future studies using hyperspectral reflectance to achieve accurate monitoring of these ecosystems. While indices like PRI and CCI are performing as well as our methods using the full spectrum analysis at the canopy scale, the application of the full spectrum might be more robust for space-based measurements. In addition, we find seasonal changes of canopy
reflectance nearby the red-edge, which could be related to leaf structural changes or chlorophyll-a and b changes. Our PLSR coefficients are good references for customizing VIs to infer the photosynthetic seasonality in evergreen forest when there are restrictions to use the specific bands from currently existing VIs (such as PRI and CCI). While our current study is limited to a subalpine evergreen forest and canopy-scale measurements, applications to other regions, vegetation types and observational platforms will be a focus for future research.

*Code and data availability.*   We thank the AmeriFlux Niwot Ridge PIs for making the eddy-covariance flux data available for this study. The PROSAIL model in the Julia programming language used in our study can be obtained from https://github.com/climate-machine/ LSM-SPAM.

## Appendix A:  Bi-directional reflectance effect

The impact of geometry and small FOV are relatively negligible. First, our method only used the scans when FOV is on the
needles by setting a NDVI threshold. Second, we plotted the NDVI and NIRv against the solar geometry at each individual tree targets throughout a year. NDVI/NIRv are quite homogeneous regardless of various solar geometries as shown in the following figures.



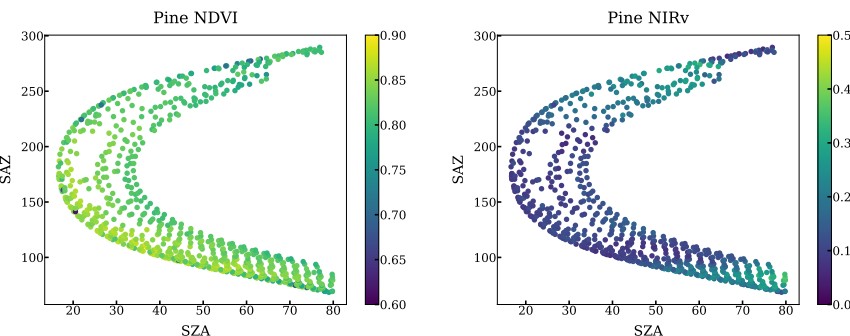

**Figure A1.** NDVI and NIRv of all scans targeting on a pine at different solar geometries throughout a year.

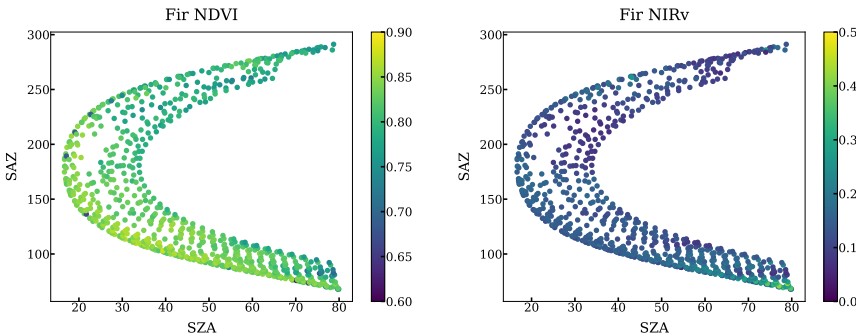

**Figure A2.** NDVI and NIRv of all scans targeting on a fir at different solar geometries throughout a year.

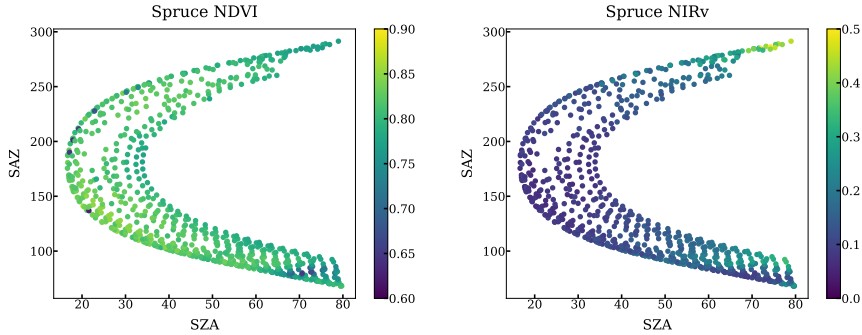

**Figure A3.** NDVI and NIRv of all scans targeting on a spruce at different solar geometries throughout a year.





## Appendix B: Detailed processes on integrating daily-averaged canopy reflectance

First, we chose scans targeting vegetation only by requiring an NDVI greater than 0.6. Second, it is important to ensure that
the solar irradiation did not change between the acquisition of the solar irradiance and the reflected radiance measurement. To achieve this, we matched the timestamps of a PAR sensor (LI-COR LI-190SA, LI-COR Environmental, Lincoln, Nebraska, US) to the timestamps of PhotoSpec, and compared the PAR value from the PAR sensor during the PhotoSpec irradiance acquisition with PAR during the actual target scan of the reflected radiance from vegetation. We only used the scans when the ratio of the two was $1.0\pm0.1$, ensuring stable PAR conditions. Third, in order to avoid unstable PAR because of clouds (Dye,
2004), we also removed cloudy scenes by requiring PAR to be at least 60% of a theoretical maximum driven by solar geometry (Fig. B1). Further, only data when PAR was greater than 100 $\mu$mol $m^{-2}s^{-1}$ were considered to eliminate the impact of low solar angles on reflectance data. The VIs shown in Fig. 5 were extracted in the same fashion as above.

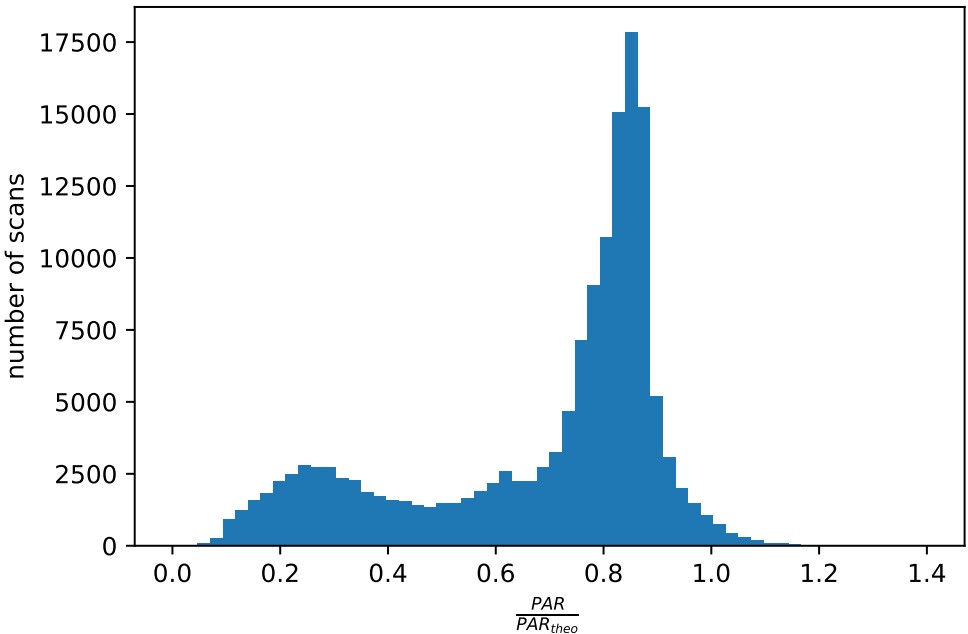

**Figure B1.** The distribution of ratio PAR to theoretical PAR from all individual scans.

## Appendix C: PROSAIL fits

We used the following range constraints for variables included in the state vector of PROSAIL inversion:

– Leaf mesophyll structure (N): 0.9–1.1

     – Chlorophyll content (Cchl): 0–120 $\mu$mol cm$^{-2}$





- Carotenoid content(Car): 0–70 $\mu\mathrm{mol\,cm^{-2}}$

- Anthocyanin content(Ant): 0–10 $\mu\mathrm{mol\,cm^{-2}}$

- Brown pigments(Cbrown): 0–0.6

- Water content (Cw): 0–0.2 cm

- Dry matter content (Cm): 0–0.2 $\mathrm{g\,cm^{-2}}$

- Xantophyll cycle status (Cx) 0–1

- Leaf area index (LAI): fixed to 4.2

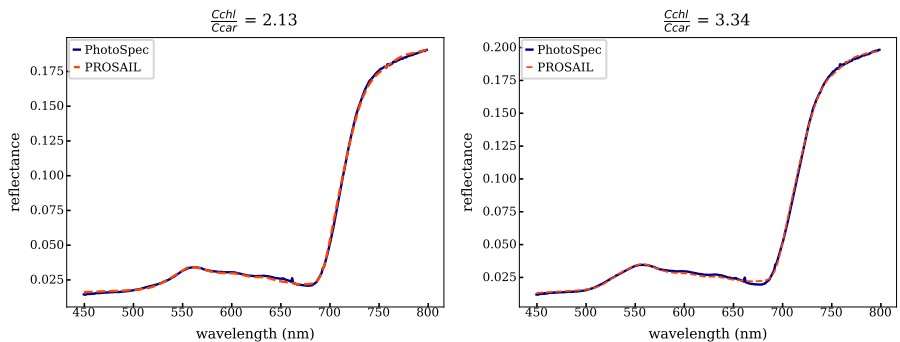

**Figure C1.** The observed and fitted reflectance spectra at low (left) and high (right) $\frac{Cchl}{Ccar}$

**Appendix D:  Sigmoid fit**

The sigmoid equation is:

$$y = b + \frac{a - b}{1 + exp(\frac{d-x}{c})}$$

In this form, a and b represent the maximum and minimum values of the sigmoid fit. And d is the half maximum of the fit. We obtained the optimal values of these parameters.

Proof:

If $x \to +\infty$, $exp(\frac{d-x}{c}) \to 0$. So,

$$\lim_{x \to +\infty} y = a$$





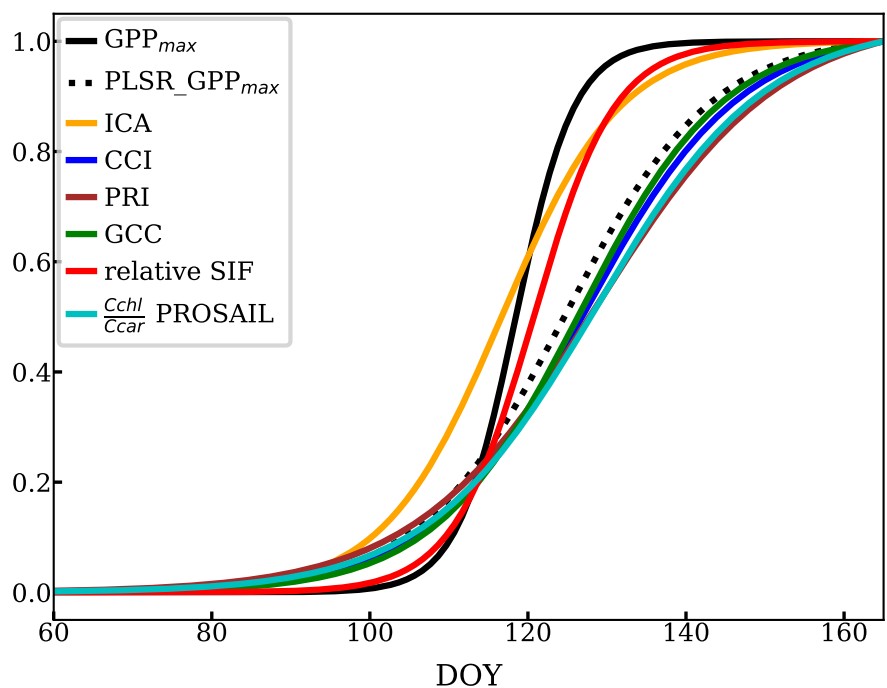

**Figure D1.** The sigmoid fits of the onset of growth from different methods and more VIs. The fits are scaled to the same range for the purpose of showing.

If $x \rightarrow -\infty$, $exp(\frac{d-x}{c}) \rightarrow +\infty$. So,

$$\lim_{x \rightarrow -\infty} y = b$$

The first derivative of y is

$$\frac{dy}{dx} = \frac{a-b}{(1+exp(\frac{d-x}{c}))^2} exp(\frac{d-x}{c}) \frac{1}{c}$$

At the half maximum point (x = $x_{half}$), $y = \frac{a+b}{2}$. Therefore, we need to solve:

$$\frac{a+b}{2} = b + \frac{a-b}{1+exp(\frac{d-x_{half}}{c})}$$

Hence, $x_{half}$ = d.

*Author contributions.* RC, TSM, DD, and CF designed research. RC, TSM, DD, DRB, BAL, SPB, PDB, KG, SL, ADR, JS, and CF performed data analyses. RC, TSM, DD, DRB, BAL, SPB, PDB, KG, SL, ADR, JS, and CF wrote the paper.





*Competing interests.* I declare that neither I nor my co-authors have any competing interests.

*Acknowledgements.* We thank the sponsors from the Caltech Graduate First-year Fellowship, NASA Carbon Monitoring Systems program
(Award NNX16AP33G) to D. R. Bowling. The US-NR1 AmeriFlux site has been supported by the U.S. DOE, Office of Science through the
AmeriFlux Management Project (AMP) at Lawrence Berkeley National Laboratory under Award Number 7094866. The National Center for
Atmospheric Research (NCAR) is sponsored by NSF.



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
