# Peer review of "Decomposing reflectance spectra to track gross primary production in a subalpine evergreen forest"

_Biogeosciences, 2020_

## Referee Comment (RC1) · Anonymous Referee #1 · 26 Mar 2020

General comments: The manuscript "Decomposing reflectance spectra to track gross primary production in a subalpine evergreen forest" aims to investigate the link between seasonal changes in the canopy reflectance (400-900 nm) of a boreal forest and the GPP changes, measured from flux tower measurements. To do so, the authors apply a technique for decomposing the reflectance into independent components (ICA) and derive a PLSR-based factor for explaining the link with the parameter "LUEs/GPPmax".

Although the manuscript contains several interesting elements, a clear hypothesis is missing (including novel research questions) and several definitions and underlying mechanisms should be better explained. For example, the authors are interested in

the red-edge region where the chlorophylls absorb but don't present a clear strategy for detecting chlorophyll pigment changes (although they are later retrieved by inversion). It is well known that the Car/Chl ratio is the main driver of photosynthetic behaviour on a seasonal scale (L59-61), i.e. altering the ratio between energy dissipation and energy harvesting. Hence, on a seasonal scale the spectral variability would be expected to occur in the pigment absorption regions of those pigments. The authors should highlight which information can be potentially provided by their technique and how it improves (?) the tracking of GPP compared to the standardly used methods (e.g. VIs).

Further, the authors aim to evaluate the pigment driven spectral changes (where, when and why). In this regard the authors could further highlight the seasonal dynamics of the detected components in respect to the spring recovery in boreal forests. Does it provide more info compared to the VI dynamics?

Finally, there are several jumps in the storyline, use of unclear terminology/method descriptions (L141-143, L190-194) and missing parameters definitions (L187). The presentation of the results is sometimes fragmented (L183-185) or not clear from the graphs (L217-L218, GPPmax is not shown). All these aspects need to be thoroughly reviewed before acceptance of the manuscript.

Specific comments From L43-48 it could be misunderstood that LUE of decidious forests is not affected by biotic factors, while LUE changes due to e.g. pigment composition occur in combination with structural changes, which in fact you can also term a "biotic" factor. The term "biotic" refers to higher-level ecosystem interactions and is less appropriate in the LUE-photosynthesis terminology here. Please rephrase. What is the link with the "differentiation in NPQ pathways" and SIF, which are suddenly mentioned at L75. Is this relevant for seasonal patterns/this manuscript? L77: you are comparing fluorescence radiance with reflectance, which varies strongly in the 400-900 region and is moreover a ratio, not a radiance to compare SIF with. L215-216: why would low PAR not drive photosynthesis? Please reformulate this sentence, pointing to the controlling factors in winter/spring. L78-81: The mechanisms are not clearly explained

here. What about the seasonal radiance budget, i.e. the "abiotic" factors?

Methodology After filtering the data based on light conditions and snow, how many winter days actually remain? Please mention the amount of samples, for both winter and growing season, also in Fig. 1. Relative SIF: please elaborate on how the normalization is done (raw data, wavelength range). Since you argued in the introduction that the structural changes are less an issue for coniferous forest, what is the true (or expected) impact of this normalization for SIF? What is the difference with not normalizing? Did you quantify this? LUElightL/LUEtotal: these are supposingly daily values? How APAR was defined/calculated based on the raw data and show a plot of the methodology described in L160. Moreover these parameters are not clearly presented later on and Fig.2 does not give a sufficient visual on the calculation/importance of these parameters. Are they relevant for the story? L155: It is claimed that PAR levels between 1000 and 1500 mol m-2 s-1 are reached throughout the whole year, but that is not what is seen from Fig. 3, showing PAR values hardly exceeding 1000 mol m-2 s-1. LUEs: this parameter suddenly appears at L187, without any previous definition! Also, what does the reader need to understand from the LUEs/GPPmax parameter? Please, elaborate the choice of this parameter and how it should be interpreted in terms of vegetation dynamics. Pigment contents: is there a reason why Chlorophyll content is lacking? This does not follow the line of the objectives. L190: rephrase this sentence for a better understanding of the final aim. The resulting coefficient is given somewhere or expected later in the results? Which four PLSR components are you referring to? L203: the raw input reflectance data is unclear here. Also, please further highlight which pigments you are inverting from the reflectance and why.

Results Section 3.1: this whole section refers to results about GPP max without clearly referring to results on this parameter. Please refer better to the results shown in Fig. 3 and check why LUElighL and LUEtotal are not shown in the graph (but mentioned in the legend). L226: please refer first to the observations in the figure in the main text, and for further details refer in addition to the supplementary figures. Fig. 4: "Annual

mean reflectance": correct this as the "Annual mean log scaled R" Section 3.2: The link between the seasonality of the spectral components and GPP max seems interesting, but there is a clear difference in the onset of the components 1 and 2 (the more dynamic ones) which might be in addition highlighted and of scientific interest. Section 3.3: The explanation of the methodology in this section needs to be improved. Please be more concrete in terminology (L278: transition period, noise) and what exactly you are referring to. L279-L281: What do you mean with that the high-frequency variations are not captured by any method? The PRI captures the variation of the most dominant feature in the PLSR coefficients. So why would these variations not be related to pigment content? Section 3.4: L302: Please refer to the graphs.

---

## Author Comment (AC1) · 13 Apr 2020

**Detailed response to referee 1's comments and suggestions**

April 13, 2020

General comments: The manuscript "Decomposing reflectance spectra to track gross primary production in a subalpine evergreen forest" aims to investigate the link between seasonal changes in the canopy reflectance (400-900 nm) of a boreal forest and the GPP changes, measured from flux tower measurements. To do so, the authors apply a technique for decomposing the reflectance into independent components (ICA) and derive a PLSR-based factor for explaining the link with the parameter "LUEs/GPPmax".

We thank you for reviewing our work. Please find a point-by-point response below. All the changes will be reflected in the revised draft.

Although the manuscript contains several interesting elements, a clear hypothesis is missing (including novel research questions) and several definitions and underlying mechanisms should be better explained.

Thank you for pointing out the lack of clarity in our hypothesis. We will make sure to clarify our hypothesis in the revised draft. In short, we hypothesized that measuring

hyperspectral reflectance at the canopy level is able to track the Light Use Efficiency (LUE) at a sub-alpine evergreen forest. For this, we used reflectance as a proxy for the contents of photosynthetic/photoprotective pigments, which was then linked to the photosynthetic LUE.

The first novelty in our study is we continuously measured hyperspectral reflectance at the canopy level. In previous studies, canopy level reflectance was either only simulated with radiative models or observed sparsely and mostly performed at discrete broad spectral bands. Coincident with our year-long reflectance measurements, pigments were sampled across the canopy so that we could track the onset and cessation of photosynthesis, and seek to provide a direct link between changes in canopy reflectance and pigment contents at the canopy scale.

The second novelty is a comprehensive scheme to link the seasonality of photosynthesis at the canopy scale to photoprotective pigments. This exploratory scheme includes empirical methods as well as process-based analysis. Previous studies have used one of the two methods. However, the availability of reflectance observation, pigment samples, and flux measurements allowed us to test our hypothesis both empirically and physically.

Additionally, the PhotoSpec system we used also measures Solar-Induced Fluorescence (SIF), which was shown to track the seasonality of photosynthetic LUE from previous studies. Thus, we included the SIF analysis to our work in order to highlight the different de-excitation pathways of excited chlorophylls by photoprotective pigments and SIF.

For example, the authors are interested in the red-edge region where the chlorophylls absorb but don't present a clear strategy for detecting chlorophyll pigment changes (although they are later retrieved by inversion).

Thank you for asking this question. We will add the measurement of chlorophyll content in the revised section 2.4. We measured the chlorophyll content along with xanthophyll

content and carotenoid content. The chlorophyll content was used in the calculation of the car/chl ratio and the comparison against the PROSAIL inversion results. In fact, there are changes around the red edge but the absence of clear changes in the peak Chl absorption regions points to small Chl changes throughout the season. However, these changes around the red edge could be related to Chl-a and Chl-b or structural changes.

It is well known that the Car/Chl ratio is the main driver of photosynthetic behaviour on a seasonal scale (L59-61), i.e. altering the ratio between energy dissipation and energy harvesting. Hence, on a seasonal scale the spectral variability would be expected to occur in the pigment absorption regions of those pigments. The authors should highlight which information can be potentially provided by their technique and how it improves (?) the tracking of GPP compared to the standardly used methods (e.g. VIs).

Our empirical methods showed and agreed on the spectral features in the reflectance were attributed to the pigments which are responsible for the photosynthetic seasonality, which was further validated by the process-based analysis using PROSAIL inversion. We will further highlight the link among car/chl, spectral features in the reflectance, and LUE in our revised manuscript.

Although we showed the performance of our empirical methods and well-established Vegetation Indices (VIs), our goal was not to achieve a higher correlation coefficient from our method than the VIs. Instead, our work focuses on mechanistically explaining where, when, and why certain wavelength regions are sensitive to the canopy LUE, which validated the high/low correlation coefficients from the VIs.

Further, the authors aim to evaluate the pigment driven spectral changes (where, when and why). In this regard the authors could further highlight the seasonal dynamics of the detected components in respect to the spring recovery in boreal forests. Does it

provide more info compared to the VI dynamics?

Thanks again for pointing out our unclear discussion on comparing our methods with VIs. We will strengthen our discussion in the revision. In section 3.5 comparison across methods, what we really wanted to highlight is the difference between the reflectance change driven by the pigments and SIF. Since the spectral shape and CCI (the representatives of other VIs) both related to the chl:car ratio, the same behavior from the spectral change and CCI are, in fact, expected in Figure 9. Interestingly, the PLSR and ICA decompositions didn't significantly improve (or provide more info) to the ability of existing VIs. This is important because it validates the idea that 'more simple' approaches might be sufficient for tracking the seasonality in evergreen systems.

Finally, there are several jumps in the storyline, use of unclear terminology/method descriptions (L141-143, L190-194) and missing parameters definitions (L187). The presentation of the results is sometimes fragmented (L183-185) or not clear from the graphs (L217-L218, GPPmax is not shown). All these aspects need to be thoroughly reviewed before acceptance of the manuscript.

Thank you for the comment. In the revision, we will correct the definitions, keep the consistency of terminology, and make sure the plots are explanatory to the text.

Specific comments From L43-48 it could be misunderstood that LUE of decidious forests is not affected by biotic factors, while LUE changes due to e.g. pigment composition occur in combination with structural changes, which in fact you can also term a "biotic" factor. The term "biotic" refers to higher-level ecosystem interactions and is less appropriate in the LUE-photosynthesis terminology here. Please rephrase.

Thanks for this comment. We will rephrase carefully in the revised manuscript. We will refer to changes in pigment compositions as 'needle biochemistry.'

What is the link with the "differentiation in NPQ pathways" and SIF, which are suddenly mentioned at L75. Is this relevant for seasonal patterns/this manuscript?

We will add transitional introductions to the revised manuscript and explain the necessity of the comparison of SIF and reflectance spectra. SIF has been shown to track the photosynthetic seasonality in previous studies. However, SIF and the spectral change in reflectance represent different de-excitation pathways of excited chlorophyll. Photo-Spec measures both hyperspectral reflectance and SIF, which enables us to compare the seasonality captured by these pathways.

L77: you are comparing fluorescence radiance with reflectance, which varies strongly in the 400-900 region and is moreover a ratio, not a radiance to compare SIF with.

In the analysis, we used relative SIF, which is SIF normalized by the reflected near-infrared radiation. In the revision, we will rephrase the paragraph and introduce the relative SIF in this section. Relative SIF is used to account for sunlit/shaded fraction within our observation FOV, since it provides an indicator of how 'bright' the area of interest is.

L215-216: why would low PAR not drive photosynthesis? Please reformulate this sentence, pointing to the controlling factors in winter/spring.

Thanks. The original sentence was trying to inform the readers that GPP was near zero in winter, although the PAR level in winter. This matches with the strong seasonality of our LUE measurement, GPPmax. And figure S3 shows that instead of PAR, Tair is one of the controlling factors in winter. We will rephrase it in the revision to make it clearer.

L78-81: The mechanisms are not clearly explained here. What about the seasonal radiance budget, i.e. the "abiotic" factors?

Thank you for pointing out the missing part of SIF. In the revised draft, we will elaborate on the explanation of SIF mechanisms and introduce relative SIF. While there is still enough light to drive photochemistry in winter, frozen soils and boles limit water transport as needles must dissipate excess energy as heat. The primary mechanism for increased NPQ is through sustained energy dissipation by photoprotective pigments which co-varies seasonally with SIF (Magney et al., 2019).

Methodology After filtering the data based on light conditions and snow, how many winter days actually remain? Please mention the amount of samples, for both winter and growing season, also in Fig. 1.

We have 96 days of spectral samples between DOY 100-300 and 115 days of spectral samples in the rest of 165 "winter" days.

In the pigment analysis, we have 6 days of both spectral and pigment samples between DOY 100-300 and 4 days in the rest of 165 "winter" days.

In figure 1, there are 39 days in the growing season and 113 days in the dormancy.

We will include these statistics in the revision.

Relative SIF: please elaborate on how the normalization is done (raw data, wavelength range). Since you argued in the introduction that the structural changes are less an issue for coniferous forest, what is the true (or expected) impact of this normalization for SIF? What is the difference with not normalizing? Did you quantify this?

Relative SIF was SIF normalized by the reflected near-infrared radiance at 755nm. This normalization will make SIF more comparable to a 'SIF yield', as it is a ratio effectively correcting for incoming irradiance, and sunlit/shaded fraction (see above). The attached fig.1 is similar as we did in Figure 5d but with SIF and relative SIF. The seasonal cycles of relative SIF and SIF are well correlated. Relative SIF is more cor-

related with the GPPmax in seasonal variations. However, the sub-seasonal change in the growing season is captured more by relative SIF.

LUElightL/LUEtotal: these are supposingly daily values? How APAR was defined/calculated based on the raw data and show a plot of the methodology described in L160. Moreover these parameters are not clearly presented later on and Fig.2 does not give a sufficient visual on the calculation/importance of these parameters. Are they relevant for the story?

GPPmax, LUElightL, and LUEtotal are light use efficiencies (LUE) at different abiotic status: 1) GPPmax: light-saturated LUE; 2) LUElightL: light-limited LUE; 3) LUEtotal: mean status LUE. They are all calculated as daily values. To make the discussion concise and clear in the revised draft, we will only show the results of GPPmax to represent LUE in the main text as it is more representative than LUElightL and more physiology-driven than LUEtotal. We will make sure the terminology of using LUE and GPPmax is consistent through the draft. We will also add sections on how APAR, LUElightL, and LUEtotal were calculated in the appendix with visualized explanations.

APAR was calculated from seven pairs of PAR sensors installed. One pair of sensors was installed above on the same tower where PhotoSpec is located (measuring incoming PAR and reflected PAR). The other six pairs of sensors were installed below the canopy (measuring reflected and transmitted par). The derivation of APAR is shown in the attached fig.2.

The attached fig.3 is a demonstration of how LUElightL and LUEtotal were calculated. Given a day (DOY =278 as an example), we selected the GPP measurements when the PAR level is between 100-500 $\mu$mol m$^{-2}$ s$^{-1}$. Then, we did a linear regression of those GPP measurements with their APAR levels (the cyan dots and dashed line). The slope of this regression is LUElightL. On the same day, all the GPP measurements that happened when the PAR level is above 100 $\mu$mol m$^{-2}$ s$^{-1}$ are the orange crosses in

the plot. We calculate the ratio of GPP and APAR of those orange points, and the daily mean of the ratio is the LUEtotal.

L155: It is claimed that PAR levels between 1000 and 1500 $\mu$mol m$^{-2}$ s$^{-1}$ are reached throughout the whole year, but that is not what is seen from Fig. 3, showing PAR values hardly exceeding 1000 $\mu$mol m$^{-2}$ s$^{-1}$. LUEs: this parameter suddenly appears at L187, without any previous definition! Also, what does the reader need to understand from the LUEs/GPPmax parameter? Please, elaborate the choice of this parameter and how it should be interpreted in terms of vegetation dynamics.

In figure 3, PAR is low because it was calculated as the daily averaged PAR of above 100 $\mu$mol m$^{-2}$ s$^{-1}$. In the calculations of GPPmax, LUElightL, and LUEtotal, PAR and APAR are half-hourly data.

Thank you for catching this error. LUEs referred to LUElightL and LUEtotal in the discussion paper. And LUEs/GPPmax refers to GPP and LUEs. GPPmax, LUElightL, and LUEtotal are light use efficiencies (LUE) at different light regimes: 1) GPPmax: light-saturated LUE; 2) LUElightL: light-limited LUE; 3) LUEtotal: mean status LUE. We did all the analysis with these three parameters. The results were quite similar in terms of the seasonal cycle. As we explained in section 3.1 L220-225, GPPmax is a better proxy for LUE as it is more representative than LUElightL and physiology-driven than LUEtotal. Thus, we decided to only keep GPPmax in the main text. In the revision, we will make sure to use GPPmax consistently.

Pigment contents: is there a reason why Chlorophyll content is lacking? This does not follow the line of the objectives.

We didn't show chlorophyll content in figure 7 because Bowling et al (2018) and Magney et al (2019) have shown chlorophyll content didn't vary with the season in our

study site. However, we agree with your suggestion that it is important to include it to comprehensively discuss the importance of car and chl:car ratio to the seasonality of photosynthesis. We will redo the plot and add the discussion in the revision.

L190: rephrase this sentence for a better understanding of the final aim. The resulting coefficient is given somewhere or expected later in the results? Which four PLSR components are you referring to?

Four PLSR components we mentioned are the parameters used when we trained the PLSR algorithms. We will rephrase the paragraph to explicitly describe the PLSR coefficient and its role in our analysis. We will also rewrite the implementation of PLSR with clarified terminology.

L203: the raw input reflectance data is unclear here. Also, please further highlight which pigments you are inverting from the reflectance and why.

Thank you for comments. When we calculated the Jacobians, we used the log-scaled daily-averaged canopy-averaged reflectance. In PROSAIL inversion, we used daily-averaged canopy-averaged reflectance without log scaling. The inverted pigments are chlorophylls, carotenoids, and anthocyanin. We compared the inverted pigment contents of chlorophyll and carotenoids as well as their ratio to the measurements from needle samples. We will clarify this in the revised draft and will provide all averaged reflectance data as supporting online material.

Results Section 3.1: this whole section refers to results about GPP max without clearly referring to results on this parameter. Please refer better to the results shown in Fig. 3 and check why LUElighL and LUEtotal are not shown in the graph (but mentioned in the legend).

L226: please refer first to the observations in the figure in the main text, and for further details refer in addition to the supplementary figures.

*Thank you for catching the misleading legend. We will remove it. And we will refer to the plots in the section in the revision.*

Fig. 4: "Annual mean reflectance": correct this as the "Annual mean log scaled R"

*Thanks. We will correct it in the revision.*

Section 3.2: The link between the seasonality of the spectral components and GPP max seems interesting, but there is a clear difference in the onset of the components 1 and 2 (the more dynamic ones) which might be in addition highlighted and of scientific interest.

*Thank you for catching this. We are not sure the origin of the abrupt onset near DOY 50 in component 1. This abrupt change happens in all three components in Figure 4. We suspect the change was caused by background noise, such as snow from the ground, given that the PhotoSpec system operated normally during that period. However, we cannot identify the reason and will suggest this as mere speculation in the revisions.*

Section 3.3: The explanation of the methodology in this section needs to be improved.

*We will elaborate the implementation of PLSR analysis in the revision.*

Please be more concrete in terminology (L278: transition period, noise) and what exactly you are referring to.

[Figure]

The transition period refers to the onset and cessation of growing season shown in GPPmax. The noise refers day-to-day fluctuations in winter. GPPmax was consistently near $0\pm1\mu$mol m$^{-2}$ s$^{-1}$ before DOY 100, while PLSR reconstructed GPPmax varies from -2 to 5 $\mu$mol m$^{-2}$ s$^{-1}$ in the same period. This large variation is from the noisy input of reflectance, which can be seen in the ICA components. We will rephrase the sentence to be more specific.

L279-L281: What do you mean with that the high-frequency variations are not captured by any method? The PRI captures the variation of the most dominant feature in the PLSR coefficients. So why would these variations not be related to pigment content?

The high-frequency variations refer to the day-to-day fluctuations within growing season. Both PRI and PLSR coefficients are relatively flat during the growing season. They did not follow the day-to-day variations within the season. As both PRI and PLSR use the pigment absorption feature, their invariance suggests these variations are less relevant to pigment.

Section 3.4: L302: Please refer to the graphs.

Thank you for the comment. The sentence will refer to Figure 8(b).
* * *

**(a) SIF**

- SIF
- GPP$_{max}$

r$^2$ = 0.83

2018 : 2017

SIF ($mW^{-2}nm-1sr^{-1}$)
GPP$_{max}$ [$\mu mol\ m^{-2}s^{-1}$]
DOY

**(b) relative SIF**

- relative SIF
- GPP$_{max}$

r$^2$ = 0.91

2018 : 2017

relative SIF
GPP$_{max}$ [$\mu mol\ m^{-2}s^{-1}$]
DOY

**(c) all days, SIF vs. relative SIF**

r$^2$ = 0.90

relative SIF
SIF ($mW^{-2}nm-1sr^{-1}$)

**(d) DOY 100-300, SIF vs. relative SIF**

r$^2$ = 0.71

relative SIF
SIF ($mW^{-2}nm-1sr^{-1}$)

**(e) DOY 100-300, SIF vs. GPP$_{max}$**

r$^2$ = 0.57

GPP$_{max}$ [$\mu mol\ m^{-2}s^{-1}$]
SIF ($mW^{-2}nm-1sr^{-1}$)

**(f) DOY 100-300, relative SIF vs. GPP$_{max}$**

r$^2$ = 0.73

GPP$_{max}$ [$\mu mol\ m^{-2}s^{-1}$]
relative SIF

**Fig. 1.** Comparing SIF and relative SIF

[Figure]

**Fig. 2.** Observing and calculating APAR

DOY = 278

GPP $[\mu mol\ m^{-2}s^{-1}]$

APAR $[\mu mol\ m^{-2}s^{-1}]$

**Fig. 3.** A demonstration of calculating LUEtotal and LUElightL

---

## Referee Comment (RC2) · Anonymous Referee #2 · 18 Apr 2020

This paper presents an exploratory study of a very nice data set studying 400-900 nm reflectance, fluorescence, and GPP along with other ancillary measurements including carotenoid composition at leaf scale at the Niwot Ridge, Colorado, USA site in a subalpine evergreen forest. As I don't believe this complement of data has been presented at such a site, the data analysis is appropriate for publication in Biogeoscience. However, there are several details that must be addressed before the paper is acceptable for publication. As the first reviewer provided a number of suggestions, this reviewer will try to present a few points not already covered there. The paper contained many small errors that, while minor, led to making the paper a difficult read. Hopefully this will be fixed in a revision. Otherwise I found the paper to be very informative and interesting.

[Figure]

General comments:

I have two main issues with the paper. The first is that SIF contains a component of PAR while reflectance does not. This makes it a bit unfair to compare any of the reflectance-based quantities directly with GPPmax (similarly affected by PAR as is SIF) and also to derive the PLSR reconstruction of GPPmax without consideration of PAR. It may be fine to do the reconstruction of other quantities (related to pigments) without account of PAR, but generally not GPPmax. A much better result may come from normalizing GPPmax with respect to PAR (or daily averaged PAR or daily averaged potential PAR) and performing the reconstruction on this quantity. This is particularly important in Fig. 9. Some statements may need to be modified after taking into account PAR (e.g., paragraph starting on L. 320).

The second comment relates to the terminology around the component analysis. It would be helpful if the reflectance can be written in the form of equations explaining the decomposition. Then it may be more clear to express what is being plotted. In my understanding of the terminology, a coefficient should be a number multiplied by a particular spectral component to reconstruct a given spectrum. The term temporal component is confusing to me as this is not what has been decomposed, but rather it is the coefficient of a given spectral component to reconstruct a spectrum observed at a particular time if I have understood correctly. The labeling of Fig. 6 is particularly difficult to understand since panel (a) shows much more than PLSR coefficients. The line labeled GPP_max would be more clear if it had PLSR coefficient in the label.

Specific comments:

L. 118. VI's are normalized such that at least some of the solar geometry effects are removed. This may not be the case for reflectance in general.

L. 142. Normalized at which wavelength(s) exactly? I'm not sure I agree that normalizing by reflected radiation is going to properly "account for the complexity of signal due to canopy structure" at this particular site. Please provide more justification of this

statement.

Since it is mentioned on L. 183 that 3 components explain more than 99.99% of variance, can you state how much variance is explained by each of the components shown.

L. 261, it is mentioned that Tair and VPD covary with GPPmax. Please provide some numbers here such as correlations.

I am confused as to what is meant by "short-term" on line 262 as a few lines above it is said to be the smoothest. Some may take short-term to mean daily. In that case, we wouldn't expect it to be smooth.

Have you definitively shown here that the green band captures variations in LUE? Isn't this only inferred?

Fig. 5, something appears incorrect with the r2 value shown in panel (f).

L. 319, What exactly is meant by diurnal? The most common use of this word in my field pertains to "of or during the day" which is commonly taken to mean sub-daily.

Same paragraph: In this work, it is not definitively shown that SIF tracks seasonal or diurnal variations better than reflectance. While SIF shows slightly higher r2, the differences were not shown to be statistically significant. It is curious that CCI gives a higher r2 value with respect to GPPmax than the PLSR analysis and the PRI gives the same value as PLSR. Also the sample of points looks different in Figs. 5 and 6.

Fig. 9: There is no goodness of fit metric here relative to measurement uncertainties. To make it more clear the fits should be shown with the observations and the residuals and fit properly evaluated with standard metrics. Otherwise the differences are not convincing.

L. 334, I may have missed something but I didn't see how this feature was shown to be directly related to LUE in the paper. This may be inferred but it wasn't directly shown.

Detailed comments:
There are a number of typos that need to be fixed, for example subscripts (L. 1, L. 226, L. 234, Fig. 3, Fig. 6 panel (b)).

There are a number of statements that need to be clarified or corrected for language (see L. 2, for example, "Estimating . . . is a primary uncertainty" would be better phrased as "Estimation of...corresponds to a primary source of uncertainty" or similar). See also lines 11-12 (unclear sentence, L. 14, etc.).

Line 21: Satellites do not measure GPP, rather GPP can be inferred and usually make use of other data.

It's a little confusing in L. 159 to start with "To implement Eq. (1), then define a particular case for Eq. 1. Please rephrase.

L. 164-166. It's not clear at this point what the meteorological data are included for. L. 165 would be more clear to say that daily mean . . . were computed from . . . .

The first reviewer was unclear about what LUEs/GPPmax is. I think I figured it out but it took a lot of time and was very unclear.

Line 217. Is LUElight the same as LUElightL defined above?

L. 226: Confusion regarding Fig. 3 (not S3) but also most of which is repeated in Fig. S2 but with the lines that are referred to in Fig. 3. Suggest to include only one figure with all the lines (in the main manuscript). A similar thing happens with Fig. 9 and D1. Suggest to include only one of these.

Fig. 3: Specify that these are daily-averaged quantities?

Fig. 4 caption: Only the 2nd component shows the carotenoid Jacobian.

Sect. 3.2, first par. There is a lot of information in this paragraph. It might be more effective if it was split up.

L. 268, the word "thus" here is confusing.

Fig. 6: Panel (a) labeling is very confusing. First, the title of panel (a) as well as the label on right side is confusing as these are not really coefficients are they, they are either components of combinations of components (caption is also unclear)? The caption states that the overlaid solid line is the 2nd ICA component, but there are two solid lines. The blue line in the legend is labeled as GPPmax, but it isn't really GPPmax as labeled in the bottom. Would suggest to just remove the titles of both panels.

L. 297 should be "support".

It would be better to subscript the small letters in Cchl, Ccar, and Cant.

Fig. B1 caption should say theoretical maximum (or clear sky)
* * *

---

## Author Comment (AC2) · 7 May 2020

This paper presents an exploratory study of a very nice data set studying 400-900 nm reflectance, fluorescence, and GPP along with other ancillary measurements including carotenoid composition at leaf scale at the Niwot Ridge, Colorado, USA site in a sub-alpine evergreen forest. As I don't believe this complement of data has been presented at such a site, the data analysis is appropriate for publication in Biogeoscience. However, there are several details that must be addressed before the paper is acceptable for publication. As the first reviewer provided a number of suggestions, this reviewer will try to present a few points not already covered there. The paper contained many small

errors that, while minor, led to making the paper a difficult read. Hopefully this will be fixed in a revision. Otherwise I found the paper to be very informative and interesting.

Thank you very much for your generous comments. We appreciate you recognized the novelties in our work. We apologize for the confusion from mislabeled plots and complex phrases. We will answer your questions point-by-point. The corrected plots, as well as responses, will be reflected in an updated version of our manuscript.

General comments: I have two main issues with the paper. The first is that SIF contains a component of PAR while reflectance does not. This makes it a bit unfair to compare any of the reflectance- based quantities directly with GPPmax (similarly affected by PAR as is SIF) and also to derive the PLSR reconstruction of GPPmax without consideration of PAR. It may be fine to do the reconstruction of other quantities (related to pigments) without account of PAR, but generally not GPPmax. A much better result may come from normalizing GPPmax with respect to PAR (or daily averaged PAR or daily averaged potential PAR) and performing the reconstruction on this quantity. This is particularly important in Fig. 9. Some statements may need to be modified after taking into account PAR (e.g., paragraph starting on L. 320).

Thank you for pointing this out, we did not properly introduce relative SIF in the introduction. In Fig. 5(d) and Fig. 9, we accounted for the PAR impact on SIF by using relative SIF, which is SIF normalized by the reflected near-infrared radiance. In this way, relative SIF is more analogous to the SIF yield, which accounts for the incident irradiance on needles within our field of view. We will rewrite the introduction as well as methods on relative SIF to clarify this.

We agree with you that normalization is necessary. We used equation 1 to derive light use efficiency (LUE), which is conditioned on both PAR and fPAR. Benefiting from the available APAR measurements, we can use in situ APAR as the normalization

factor instead of PAR and presumed fPAR in previous studies. According to the light response curve of photosynthesis (Fig. 2), we summarized three different scenarios LUE: 1) light-limited (low light, LUElightL); 2) carboxylation rate-limited (high light, GPPmax); and 3) daily average (LUEtotal).

LUElightL is the fitted slope of GPP and APAR when PAR is between 100-500 $\mu mol m^{-2}s^{-1}$. The fit was forced to go through the origin as the equation has no intercept.

LUEtotal is the daily average of $\frac{GPP}{APAR}$ during the day.

Theoretically, we could calculate GPPmax similar to what we did for LUElightL, i.e. regressing GPP against APAR when PAR is saturated. Unfortunately, there is a 26-day gap in APAR measurement in the beginning of our study period. We could also normalize GPP by PAR, as you suggested. Yet, it requires assumptions about fPAR, which adds further uncertainties.

Considering GPP often asymptotes when PAR is greater than 1000 $\mu mol m^{-2}s^{-1}$, the fitted slope could be biased by the spread of data points. Thus, we defined GPPmax, the average GPP within a moderately narrow window of PAR (1000-1500 $\mu mol m^{-2}s^{-1}$), to represent the LUE when the carboxylation rate is limited. In this way, we achieved normalizing GPP without missing more data because of APAR.

Those two different definitions of GPPmax are significantly linearly correlated (in figure 1 in this response document). If GPPmax is defined as the average of normalized GPP by PAR (y-axis), the analyses on the seasonal cycle will differ little from GPPmax defined as mean GPP at high PAR (x-axis). Although GPP normalized PAR results

in the correct unit of LUE, it is easily mistaken as fPAR has been considered. To avoid this confusion, we chose to use mean GPP at PAR between 1000 and 1500 $\mu mol m^{-2} s^{-1}$.

We also tested the LUE defined in three scenarios: LUElightL, GPPmax, and LUEtotal for all the analyses in the manuscript. They behave similarly, and these results are included in the supplementary information. We considered that needles spent more time at highlight intensity in the daytime. Hence, we decided to use GPPmax as the only proxy of LUE in the main text. We kept the other two matrices to the supplementary. The labels were misleading in the main text. We will clean them up in the updated draft.

The second comment relates to the terminology around the component analysis. It would be helpful if the reflectance can be written in the form of equations explaining the decomposition. Then it may be more clear to express what is being plotted. In my understanding of the terminology, a coefficient should be a number multiplied by a particular spectral component to reconstruct a given spectrum. The term temporal component is confusing to me as this is not what has been decomposed, but rather it is the coefficient of a given spectral component to reconstruct a spectrum observed at a particular time if I have understood correctly. The labeling of Fig. 6 is particularly difficult to understand since panel (a) shows much more than PLSR coefficients. The line labeled GPPmax would be more clear if it had PLSR coefficient in the label.

Thank you for the advice. We will change "temporal components" to "temporal loadings". We will add the following expression to the draft when it is updated: ICA:

$$-log(R_{\lambda,DOY}) = \sum_i (spectral\ component^i_\lambda \cdot temporal\ loading^i_{DOY})$$

PLSR:
$$GPP_{max,DOY} = -log(R_{\lambda,DOY}) \cdot PLSR\ coefficient_\lambda$$

Specific comments: L. 118. VI's are normalized such that at least some of the solar

geometry effects are removed. This may not be the case for reflectance in general.

Thank you for pointing this out. We agree the experiment related to this sentence is not enough to clarify the sun-sensor geometry impact on reflectance. Thus, we did a PLSR analysis on individual measurements of phase angle and reflectance for three summer days (2017-7-1 to 2017-7-3), such as figure 2 in this response document. The results are similar using the measurements from other days.

Indeed, the reflectance has different sensitivities not just related to phase angle. However, the poor correlation of PLSR reconstructed phase angle and the measurement suggests the variations in phase angle are not critical to account for change in reflectance. In our manuscript, we primarily removed the bi-directional impact by averaging all the individual reflectance measurements at different solar and viewing geometries over the course of a day.

L. 142. Normalized at which wavelength(s) exactly? I'm not sure I agree that normalizing by reflected radiation is going to properly "account for the complexity of signal due to canopy structure" at this particular site. Please provide more justification of this statement.

Thanks for pointing out this misleading sentence. SIF was normalized by reflected near-infrared radiation in the retrieval window (745-756nm) to account for the sunlit/shaded fraction within our FOV. Benefiting from the small FOV (0.7°) of PhotoSpec, we think the complexity of canopy structure has minimal impact on our signal.

Since it is mentioned on L. 183 that 3 components explain more than 99.99% of variance, can you state how much variance is explained by each of the components shown.

Because ICA minimizes the dependencies of the second-order moment (variance) and higher, the randomness during the minimization makes the explained variance and order of individual components unclear (Hyvärinen and Oja, 2000). In our calculation, the ICA algorithm reduced the dimension of the input matrix by eigenvalue decomposition first, from which the first three second-order independent/orthogonal components yielded 99.99% of the variance. Then, the algorithm extracted the independent components of high-order moments from these orthogonal components.

Hyvärinen, A., Oja, E. (2000). Independent component analysis: algorithms and applications. Neural networks, 13(4-5), 411-430.

L. 261, it is mentioned that Tair and VPD covary with GPPmax. Please provide some numbers here such as correlations.

We will cross reference the figures in supplementary in the draft while also include the following the Pearson-$r^2$ value of correlations.

During the growing season:

GPPmax and VPD = 0.34,

GPPmax and Tair = 0.24,

PRI and VPD = 0.04,

PRI and Tair = 0.02,

Tair and VPD = 0.77.

The similar Pearson-$r^2$ values from the correlations with VPD and Tair are due to the dependence of VPD on Tair in the growing season.

I am confused as to what is meant by "short-term" on line 262 as a few lines above it is said to be the smoothest. Some may take short-term to mean daily. In that case, we wouldn't expect it to be smooth.

Thanks for the feedback. We meant to use "short-term" and smoothness to refer to day-to-day or sub-seasonal variations. We will revise the paragraph with a consistent description in the updated manuscript.

Have you definitively shown here that the green band captures variations in LUE? Isn't this only inferred?

Yes, we inferred the green band captured LUE (due to changes in carotenoid pigments which change absorption in the green band) by the strong performance of GCC. We will change the word "show" to "suggest".

Fig. 5, something appears incorrect with the r2 value shown in panel (f).

We found a bug in the calculation and corrected it. The correct $r^2$ values of GPPmax with CCI, PRI, GCC, relative SIF, NDVI, and NIRv are 0.85, 0.81, 0.73, 0.87, 0.06, and 0.40, respectively.

L. 319, What exactly is meant by diurnal? The most common use of this word in my field pertains to "of or during the day" which is commonly taken to mean sub-daily.

Yes, it means day-to-day/subseasonal variations here, similar to the previous comment. We will replace "diurnal" with "sub-seasonal" and make sure the wording is accurate and consistent.

Same paragraph: In this work, it is not definitively shown that SIF tracks seasonal or diurnal variations better than reflectance. While SIF shows slightly higher r2, the differences were not shown to be statistically significant. It is curious that CCI gives a higher r2 value with respect to GPPmax than the PLSR analysis and the PRI gives the same value as PLSR. Also the sample of points looks different in Figs. 5 and 6.

Thanks for the suggestion. The phrase appears to make readers think we are competing with existing methods, which is misleading. Our goal focuses on mechanistically explaining where, when, and why certain wavelength regions are sensitive to canopy LUE seasonality. Thus, a similar performance from CCI and PLSR is expected as CCI uses the most sensitive band. Although SIF represents a different process from the reflectance-based methods the good performance of relative SIF has been discussed in Magney et al., 2019, which makes Niwot Ridge an interesting site to study.

After we corrected the calculation, relative SIF and PLSR ($r^2 = 0.87$) have the highest $r^2$ compared to other indices, although not significantly. The difference in figs. 5 and 6 was because we only plotted the observed GPPmax when the reflectance is also available in fig 6, while all the observed GPPmax was plotted in fig 5. In the revised draft, these two plots will only have the scatter plotted when both GPPmax and reflectance are available.

Fig. 9: There is no goodness of fit metric here relative to measurement uncertainties. To make it more clear the fits should be shown with the observations and the residuals and fit properly evaluated with standard metrics. Otherwise the differences are not convincing.

Thank you for the suggestion. We attached a plot of individual fittings and their evaluations (figure 3 in this response document). The fitted curve has been expressed as the derivation in Appendix D. The pearson-$r^2$ and p values listed in each subplot were calculated from the correlation of observed and fitted variables. The residual was calculated as the average L2 norm of the difference between observed and fitted variables normalized by the observation, such as $\frac{1}{n}\sum_i(\frac{x-\hat{x}}{x})^2$. Because the ICA component lacks a clear sigmoid shape due, ICA has a larger residual.

L. 334, I may have missed something but I didn't see how this feature was shown to be directly related to LUE in the paper. This may be inferred but it wasn't directly shown.

We will be more precise in phrasing. The sentence will be changed to "The main spectral feature centered around 530 nm is most important for *infering* the seasonal cycle of reflectance (400 – 900 nm) and GPPmax, which corresponds to changes in carotenoid content."

Detailed comments: There are a number of typos that need to be fixed, for example subscripts (L. 1, L. 226, L. 234, Fig. 3, Fig. 6 panel (b)).

Thanks a lot for pointing out the errors. We will correct the typos and make sure all the subscripts are coded correctly.

There are a number of statements that need to be clarified or corrected for language (see L. 2, for example, "Estimating . . . is a primary uncertainty" would be better phrased as "Estimation of...corresponds to a primary source of uncertainty" or similar). See also lines 11-12 (unclear sentence, L. 14, etc.).

We will rephrase those unclear sentences and be more precise.

[Figure]

Line 21: Satellites do not measure GPP, rather GPP can be inferred and usually make use of other data.

We will be more precise in the updated draft.

It's a little confusing in L. 159 to start with "To implement Eq. (1), then define a particular case for Eq. 1. Please rephrase.

We are going to rephrase the sentence as "We followed the format of Eq. (1) to define light-limited LUE (LUElightL) as..." This paragraph will also be rephrased to make it more clear that LUElightL and LUEtotal will be presented in supplementary instead of main text.

L. 164-166. It's not clear at this point what the meteorological data are included for. L. 165 would be more clear to say that daily mean . . . were computed from . . ..

We will replace "meteorological variables" with "Air Temperature (Tair) and Vapor Pressure Deficit (VPD)" in L148. And, rephrase L164-166 as daily Tair and VPD were computed from averaging half-hourly Tair and VPD when..."

The first reviewer was unclear about what LUEs/GPPmax is. I think I figured it out but it took a lot of time and was very unclear.

Thanks for taking the time to sort it out. We apologize for the confusion. In the updated draft, we will clarify the usage of acronym LUE and be consistent with GPPmax as the only proxy for LUE in the main text. We did this to avoid confusion by presenting too many 'proxies' for canopy photosynthesis. The results for LUElightL and LUEtotal will be put in the supplementary.

Line 217. Is LUElight the same as LUElightL defined above?

Yes. Sorry about the typo.

L. 226: Confusion regarding Fig. 3 (not S3) but also most of which is repeated in Fig. S2 but with the lines that are referred to in Fig. 3. Suggest to include only one figure with all the lines (in the main manuscript). A similar thing happens with Fig. 9 and D1. Suggest to include only one of these.

We will only include Fig. 3 and Fig. 9 in the main draft, and eliminate Fig. S2 and Fig. D1.

Fig. 3: Specify that these are daily-averaged quantities?

We will specify them in both the text and captions.

Fig. 4 caption: Only the 2nd component shows the carotenoid Jacobian.

Yes, because we want to emphasize the similarity of component 1 with the chlorophyll jacobian only. We will correct the caption in Fig. 4 and make sure it is consistent with the plot.

Sect. 3.2, first par. There is a lot of information in this paragraph. It might be more effective if it was split up.

Yes, we agree with you that is too tedious to read. We will split it into paragraphs and rephrase with the equations of ICA so that it is more clear.

L. 268, the word "thus" here is confusing.

We will remove "thus" from the sentence.

Fig. 6: Panel (a) labeling is very confusing. First, the title of panel (a) as well as the label on right side is confusing as these are not really coefficients are they, they are either components of combinations of components (caption is also unclear)? The caption states that the overlaid solid line is the 2nd ICA component, but there are two solid lines. The blue line in the legend is labeled as GPPmax, but it isn't really GPPmax as labeled in the bottom. Would suggest to just remove the titles of both panels.

Thank you for pointing this out. In Fig. 6(a), we will change the label of "GPPmax" to "PLSR coefficient of GPPmax". The blue curve indeed is the coefficient when the PLSR is written in a fashion of linear model, such as:

$$GPP_{max,DOY} = -log(R_{\lambda,DOY}) \cdot PLSR\ coefficient_{\lambda}$$

However, the y-axis on the right is only for PLSR coefficient not for ICA. We will clarify that both Jacobians and ICA component were scaled in this plot.

The solid blue line is the PLSR coefficient. The solid orange line is the 2nd ICA component. We will clarify them in the revised draft. The titles will also be removed.

L. 297 should be "support".

We will correct it.

It would be better to subscript the small letters in Cchl, Ccar, and Cant.

We will subscript them in the revision

Fig. B1 caption should say theoretical maximum (or clear sky)

We will change it to "theoretical maximum" in the revision.
* * *
[Figure]

[Figure]

**Fig. 1.** Two ways to calculate GPPmax.

[Figure]

**Fig. 2.** PLSR analysis on phase angle and reflectance.

[Figure]

observed GPP$_{max}$
$$\frac{10.41-0.28}{1+exp(-0.27\cdot(x-118.46))}+0.28$$
$r^2$ = 0.95, p = 0.000
residual = 6.66

PLSR GPP$_{max}$
$$\frac{11.70-0.37}{1+exp(-0.11\cdot(x-124.34))}+0.37$$
$r^2$ = 0.99, p = 0.000
residual = 7.34

ICA
$$\frac{0.07+0.03}{1+exp(-0.13\cdot(x-116.69))}-0.03$$
$r^2$ = 0.91, p = 0.000
residual = 392.77

CCI
$$\frac{0.10+0.07}{1+exp(-0.10\cdot(x-127.25))}-0.07$$
$r^2$ = 0.99, p = 0.000
residual = 3.32

relative SIF
$$\frac{0.01-0.00}{1+exp(-0.20\cdot(x-120.73))}+0.00$$
$r^2$ = 0.96, p = 0.000
residual = 0.03

CCI from PROSAIL
$$\frac{3.59-1.58}{1+exp(-0.09\cdot(x-128.60))}+1.58$$
$r^2$ = 0.98, p = 0.000
residual = 0.00

**Fig. 3.** Individual sigmoid fits of timeseries of interest. The fitted curve are expressed in the format as appendix D. The pearson-r2 and p values are for the observed and fitted variables.

---

## Author Response (AR1)

**General Response to all Referee Comments**

Many thanks to the referees for detailed and constructive reviews of the manuscript. The comments have greatly helped us to improve our submission.

The referees raised following major issues:
1) lack of a clear introduction on the hypothesis/novelty/significance of this study
2) the misconceptions caused by using multiple definitions inconsistently on Light Use Efficiency (LUE)
3) unclear explanation on relative SIF
4) unnecessary and erroneous elements in the plots
5) some figures were not properly referenced in the text.

We have addressed all of the issues in the responses to individual referees and in the revised manuscript.

In general,

1) We have clarified that the central goal of our study is to mechanistically explain where, when, and why certain wavelength regions are sensitive to the canopy LUE, instead of only comparing the performance of any of our methods to the existing vegetation indices (VIs). To achieve this goal, we hypothesized that measuring hyperspectral reflectance at the canopy level can track the LUE at a sub-alpine evergreen forest. Uniqueness of our study lies in the integration of empirical analyses and process-based simulations using simultaneously measured reflectance, Solar-Induced Fluorescence (SIF), and pigment contents at the canopy level.

2) We now use $GPP_{max}$ as the only proxy for LUE in the main text, where the other proxies of LUE at a low light intensity and daily average have been condensed and referred to the supplementary material. The relevant data and plots have also been moved to the supplementary material.

3) We rewrote the introduction to SIF and relative SIF to explain the significance and meaning of relative SIF.

4) We corrected all captions and legends in figures. The redundant legends were removed as well.

5) We reorganized the text to include necessary references to figures. The necessary analyses in the responses to referees have been added to the supplementary material.

The detailed point-to-point responses for the referee's comments are given below and reflected in the change-tracked manuscript.

**Detailed response to referee #1's comments and suggestions**

General comments: The manuscript "Decomposing reflectance spectra to track gross primary production in a subalpine evergreen forest" aims to investigate the link between seasonal changes in the canopy reflectance (400-900 nm) of a boreal forest and the GPP changes, measured from flux tower measurements. To do so, the authors apply a technique for decomposing the reflectance into independent components (ICA) and derive a PLSR-based factor for explaining the link with the parameter "LUEs/GPPmax".

We thank you for reviewing our work. Please find a point-by-point response below. All the changes have been reflected in the revised draft. The line numbers mentioned correspond to the revised manuscript with tracked changes.

Although the manuscript contains several interesting elements, a clear hypothesis is missing (including novel research questions) and several definitions and underlying mechanisms should be better explained.

Thank you for pointing out the lack of clarity in our hypothesis. We have made sure to clarify our hypothesis in the revised draft. The changes have been made accordingly in the abstract and introduction. In summary, we hypothesized that measuring hyperspectral reflectance at the canopy level is able to track the Light Use Efficiency (LUE) at a sub-alpine evergreen forest. For this, we used reflectance as a proxy for the contents of photosynthetic/photoprotective pigments, which was then linked to the photosynthetic LUE.

The first novelty in our study is that we continuously measured hyperspectral reflectance at the canopy level (L68-69). In previous studies, canopy level reflectance was either only simulated with radiative models or observed sparsely and mostly performed at discrete broad spectral bands. Coincident with our year-long reflectance measurements, pigments were sampled across the canopy so that we could track the onset and cessation of photosynthesis and seek to provide a direct link between changes in canopy reflectance and pigment contents at the canopy scale.

The second novelty is a comprehensive scheme to link the seasonality of photosynthesis at the canopy scale to photoprotective pigments (L66-67). This exploratory scheme includes empirical methods as well as process-based analysis. Previous studies have used

one of the two methods. However, the availability of reflectance observation, pigment samples, and flux measurements allowed us to test our hypothesis both empirically and physically.

Additionally, the PhotoSpec system we used also measures Solar-Induced-Fluorescence (SIF), which was shown to track the seasonality of photosynthetic LUE from previous studies. Thus, we included the SIF analysis to our work in order to highlight the different de-excitation pathways of excited chlorophylls by photoprotective pigments and SIF (L87-90).

For example, the authors are interested in the red-edge region where the chlorophylls absorb but don't present a clear strategy for detecting chlorophyll pigment changes (although they are later retrieved by inversion).

Thank you for asking this question. We have added the measurement of chlorophyll content in section 2.4 (L183). We measured the chlorophyll content along with xanthophyll content and carotenoid content. The chlorophyll content was used in the calculation of the car/chl ratio and the comparison against the PROSAIL inversion results. In fact, there are some changes around the red edge but the absence of clear changes in the peak Chl absorption regions points to small Chl changes throughout the season. However, these changes around the red edge could be related to Chl-a and Chl-b or structural changes. This is still an open question.

It is well known that the Car/Chl ratio is the main driver of photosynthetic behavior on a seasonal scale (L59-61), i.e. altering the ratio between energy dissipation and energy harvesting. Hence, on a seasonal scale the spectral variability would be expected to occur in the pigment absorption regions of those pigments. The authors should highlight which information can be potentially provided by their technique and how it improves (?) the tracking of GPP compared to the standardly used methods (e.g. VIs).

Our empirical methods showed and agreed on the spectral features in the reflectance were attributed to the pigments which are responsible for the photosynthetic seasonality, which was further validated by the process-based analysis using PROSAIL inversion. We have further highlighted the link among car/chl, spectral features in the reflectance, and LUE in section 3.3 (L326-331).

Although we showed the performance of our empirical methods and well-established Vegetation Indices (VIs), our goal was not to achieve a higher correlation coefficient from our method than the VIs. We have rephrased the misleading sentences (L10, L69-72), Instead, our work focuses on mechanistically explaining where, when, and why certain wavelength regions are sensitive to the canopy LUE, which validated the high/low correlation coefficients from the VIs (L63-66).

Further, the authors aim to evaluate the pigment driven spectral changes (where, when

and why). In this regard the authors could further highlight the seasonal dynamics of the detected components in respect to the spring recovery in boreal forests. Does it provide more info compared to the VI dynamics?

> Thanks again for pointing out our unclear discussion on comparing our methods with VIs. We strengthened our discussion in the revision (L46-53, L353-356). In section 3.5 comparison across methods, what we really wanted to highlight is the difference between the reflectance change driven by the pigments and SIF. Since the spectral shape and CCI (the representatives of other VIs) both related to the chl:car ratio, the same behavior from the spectral change and CCI are, in fact, expected in Figure 9. Interestingly, the PLSR and ICA decompositions didn't significantly improve (or provide more info) to the ability of existing VIs. This is important because it validates the idea that 'simpler' approaches might be sufficient for tracking the seasonality in evergreen systems.

Finally, there are several jumps in the storyline, use of unclear terminology/method descriptions (L141-143, L190-194) and missing parameters definitions (L187). The presentation of the results is sometimes fragmented (L183-185) or not clear from the graphs (L217-L218, GPPmax is not shown). All these aspects need to be thoroughly reviewed before acceptance of the manuscript.

> Thank you for the comment. In the revision, we have corrected the definitions, kept the consistency of terminology, and made sure the plots are explanatory to the text. Please see the change tracked revision for details. Some of the changes are made in L163-166, L226-229, L216-219, L209-L212, and L243-253 with respect to the examples that the referee mentioned.

Specific comments From L43-48 it could be misunderstood that LUE of deciduous forests is not affected by biotic factors, while LUE changes due to e.g. pigment composition occur in combination with structural changes, which in fact you can also term a "biotic" factor. The term "biotic" refers to higher-level ecosystem interactions and is less appropriate in the LUE-photosynthesis terminology here. Please rephrase.

> Thanks for this comment. We have referred to changes in pigment compositions as 'needle biochemistry' (L47).

What is the link with the "differentiation in NPQ pathways" and SIF, which are suddenly mentioned at L75. Is this relevant for seasonal patterns/this manuscript?

> We have rewritten this section in the revised manuscript and explained the necessity of the comparison of SIF and reflectance spectra (L46-53, L82-90), where the definition of NPQ has been carefully rephrased. SIF has been shown to track the photosynthetic seasonality in previous studies. However, SIF and the spectral change in reflectance represent different de-excitation pathways of excited chlorophyll. PhotoSpec measures

both hyperspectral reflectance and SIF, which enables us to compare the seasonality captured by these pathways.

L77: you are comparing fluorescence radiance with reflectance, which varies strongly in the 400-900 region and is moreover a ratio, not a radiance to compare SIF with.

In the analysis, we used relative SIF, which is SIF normalized by the reflected near-infrared radiation. In the revision, we have rephrased the paragraph and introduced the relative SIF in this section (L82-90). Relative SIF is used to account for sunlit/shaded fraction within our observation FOV, since it provides an indicator of how 'bright' the area of interest is.

L215-216: why would low PAR not drive photosynthesis? Please reformulate this sentence, pointing to the controlling factors in winter/spring.

Thanks. The original sentence was trying to inform the readers that GPP was near zero in winter, although the PAR level in winter is not. This matches with the strong seasonality of our LUE measurement, $GPP_{max}$. And figure S3 shows that instead of PAR, $T_{air}$ is one of the controlling factors in winter. We have rephrased it in the revision (L239-242) to make it clearer.

L78-81: The mechanisms are not clearly explained here. What about the seasonal radiance budget, i.e. the "abiotic" factors?

Thank you for pointing out the missing part of SIF. In the revised draft, we have elaborated on the explanation of SIF mechanisms and introduced relative SIF (L52-53, 82-90). While there is still enough light to drive photochemistry in winter, frozen boles limit water transport as needles must dissipate excess energy as heat. The primary mechanism for increased NPQ is through sustained energy dissipation by photoprotective pigments which co-varies seasonally with SIF (Magney et al., 2019).

Methodology After filtering the data based on light conditions and snow, how many winter days actually remain? Please mention the amount of samples, for both winter and growing season, also in Fig. 1.

We have 96 days of spectral samples between DOY 100-300 and 115 days of spectral samples in the rest of 165 "winter" days.

In the pigment analysis, we have 6 days of both spectral and pigment samples between DOY 100-300 and 4 days in the rest of 165 "winter" days.

In figure 1, there are 39 days in the growing season and 113 days in the dormancy.

These statistics have been included in the revision (L132 and Fig. 1).

Relative SIF: please elaborate on how the normalization is done (raw data, wavelength range). Since you argued in the introduction that the structural changes are less an issue for coniferous forest, what is the true (or expected) impact of this normalization for SIF? What is the difference with not normalizing? Did you quantify this?

We have added the description of relative SIF in L152-155. Relative SIF is SIF normalized by the reflected near-infrared radiance at 755 nm. This normalization will make SIF more comparable to a 'SIF yield', as it is a ratio effectively correcting for incoming irradiance, and sunlit/shaded fraction (see above). It also eliminates most of the directional effects within the canopy. The attached plot (added to supplementary material) is similar as we did in Figure 5d but with SIF and relative SIF. The seasonal cycles of relative SIF and SIF are well correlated. Relative SIF is more correlated with the GPP$_{max}$ in seasonal variations. However, the sub-seasonal change in the growing season is captured more by relative SIF.

[Figure]

LUElightL/LUEtotal: these are supposingly daily values? How APAR was defined/calculated based on the raw data and show a plot of the methodology described in L160. Moreover these parameters are not clearly presented later on and Fig.2 does

not give a sufficient visual on the calculation/importance of these parameters. Are they relevant for the story?

GPP$_{max}$, LUE$_{lightL}$, and LUE$_{total}$ are light use efficiencies (LUE) at different abiotic status: 1) GPP$_{max}$: light-saturated LUE; 2) LUE$_{lightL}$: light-limited LUE; 3) LUE$_{total}$: mean status LUE. They are all calculated as daily values. To make the discussion concise and clear in the revised draft, we only showed the results of GPP$_{max}$ to represent LUE in the main text as it is more representative than LUE$_{lightL}$ and more physiology-driven than LUE$_{total}$. We have made sure the terminology of using LUE and GPP$_{max}$ is consistent throughout the draft. We also added sections on how APAR, LUE$_{lightL}$, and LUE$_{total}$ were calculated in the appendix with visualized explanations.

APAR was calculated from seven pairs of PAR sensors installed. One pair of sensors was installed above the canopy on the same tower where PhotoSpec is located (measuring incoming PAR and reflected PAR). The other six pairs of sensors were installed below the canopy (measuring reflected and transmitted par). The derivation of APAR is shown in the following graph.

[Figure]

Here is a demonstration of how LUE$_{lightL}$ and LUE$_{total}$ were calculated. Given a day (DOY =278 as an example), we selected the GPP measurements when the PAR level is between 100-500 µmol m-2 s-1. Then, we did a linear regression of those GPP measurements with their APAR levels (the cyan dots and dashed line). The slope of this regression is LUE$_{lightL}$. On the same day, all the GPP measurements that happened when the PAR level is above 100 µmol m-2 s-1 are the orange crosses in the plot. We calculate the ratio of GPP and APAR of those orange points, and the daily mean of the ratio is the LUE$_{total}$.

[Figure]

L155: It is claimed that PAR levels between 1000 and 1500 µmol m-2 s-1 are reached throughout the whole year, but that is not what is seen from Fig. 3, showing PAR values hardly exceeding 1000 mol m-2 s-1. LUEs: this parameter suddenly appears at L187, without any previous definition! Also, what does the reader need to understand from the LUEs/GPPmax parameter? Please, elaborate the choice of this parameter and how it should be interpreted in terms of vegetation dynamics.

In Fig.3, PAR is low because it was calculated as the daily averaged PAR of above 100 µmol m-2 s-1. In the calculations of GPPmax, LUElightL, and LUEtotal, PAR and APAR are half-hourly data.

Thank you for catching this error. LUEs referred to LUElightL and LUEtotal in the discussion paper. And LUEs/GPPmax refers to GPP and LUEs. GPPmax, LUElightL, and LUEtotal are light use efficiencies (LUE) at different light regimes: 1) GPPmax: light-saturated LUE; 2) LUElightL: light-limited LUE; 3) LUEtotal: mean status LUE. We did all the analysis with these three parameters. The results were quite similar in terms of the seasonal cycle. As we explained in section 3.1 (L168-171 and section 2 in the supplementary material), GPPmax is a better proxy for LUE as it is more representative than LUElightL and physiology-driven than LUEtotal. Thus, we decided to only keep GPPmax in the main text. In the revision, we made sure to use GPPmax consistently.

Pigment contents: is there a reason why Chlorophyll content is lacking? This does not follow the line of the objectives.

We didn't show chlorophyll content in figure 7 because Bowling et al (2018) and Magney et al (2019) have shown chlorophyll content didn't vary with the season in our study site. However, we agree with your suggestion that it is important to include it to comprehensively discuss the importance of car and chl:car ratio to the seasonality of

photosynthesis. We have discussed the measurement of chl (L183, L216-218, L339-341) in the revision.

L190: rephrase this sentence for a better understanding of the final aim. The resulting coefficient is given somewhere or expected later in the results? Which four PLSR components are you referring to?

Four PLSR components we mentioned are the parameters used when we trained the PLSR algorithms. We have rephrased the paragraph to explicitly describe the PLSR coefficient and its role in our analysis with mathematical expressions and clarified terminology (L204-221).

L203: the raw input reflectance data is unclear here. Also, please further highlight which pigments you are inverting from the reflectance and why.

Thank you for comments. When we calculated the Jacobians, we used the log-scaled daily-averaged reflectance from PhotoSpec. In PROSAIL inversion, we used the daily-averaged reflectance without log scaling. The inverted pigments are chlorophylls, carotenoids, and anthocyanin. We compared the inverted pigment contents of chlorophyll and carotenoids as well as their ratio to the measurements from needle samples. We clarified this in the revised draft (L230-231) and provided all averaged reflectance data as supporting online material.

Results Section 3.1: this whole section refers to results about GPP max without clearly referring to results on this parameter. Please refer better to the results shown in Fig. 3 and check why LUElighL and LUEtotal are not shown in the graph (but mentioned in the legend).

L226: please refer first to the observations in the figure in the main text, and for further details refer in addition to the supplementary figures.

Thank you for catching the misleading legend (Fig. 3). We have removed it and referred to the plots in the supplementary in the revision.

Fig. 4: "Annual mean reflectance": correct this as the "Annual mean log scaled R"

Thanks. We have corrected it in the revision (Fig. 4).

Section 3.2: The link between the seasonality of the spectral components and GPP max seems interesting, but there is a clear difference in the onset of the components 1 and 2 (the more dynamic ones) which might be in addition highlighted and of scientific interest.

Thank you for catching this. We are not sure the origin of the abrupt onset near DOY 50 in component 1. This abrupt change happens in all three components in Fig. 4. We suspect the change was caused by background noise, such as snow from the ground, given that the PhotoSpec system operated normally during that period. However, we cannot identify the reason and suggested this as mere speculation in the revisions (L297-300).

Section 3.3: The explanation of the methodology in this section needs to be improved.

We have elaborated the implementation of PLSR analysis in the revision (L204-L218).

Please be more concrete in terminology (L278: transition period, noise) and what exactly you are referring to.

The transition period refers to the onset and cessation of growing season shown in $GPP_{max}$. The noise refers day-to-day fluctuations in winter. $GPP_{max}$ was consistently near $0\pm1$ $\mu mol$ $m_{-2}$ $s_{-1}$ before DOY 100, while PLSR reconstructed $GPP_{max}$ varies from -2 to 5 $\mu mol$ $m_{-2}$ $s_{-1}$ in the same period. This large variation is from the noisy input of reflectance, which can be seen in the ICA components. We rephrased the sentence to be more specific (L305-309).

L279-L281: What do you mean with that the high-frequency variations are not captured by any method? The PRI captures the variation of the most dominant feature in the PLSR coefficients. So why would these variations not be related to pigment content?

The high-frequency variations refer to the day-to-day fluctuations within growing season. Both PRI and PLSR coefficients are relatively flat during the growing season. They did not follow the day-to-day variations within the season. As both PRI and PLSR use the pigment absorption feature, their invariance suggests these variations are less relevant to pigment. The specific changes have been made in L305-309.

Section 3.4: L302: Please refer to the graphs.

Thank you for the comment. The sentence has referred to Figure 8(c) (L338-339).

**Detailed response to referee #2's comments and suggestions**

This paper presents an exploratory study of a very nice data set studying 400-900 nm reflectance, fluorescence, and GPP along with other ancillary measurements including carotenoid composition at leaf scale at the Niwot Ridge, Colorado, USA site in a sub-alpine evergreen forest. As I don't believe this complement of data has been presented

at such a site, the data analysis is appropriate for publication in Biogeoscience. However, there are several details that must be addressed before the paper is acceptable for publication. As the first reviewer provided a number of suggestions, this reviewer will try to present a few points not already covered there. The paper contained many small errors that, while minor, led to making the paper a difficult read. Hopefully this will be fixed in a revision. Otherwise I found the paper to be very informative and interesting.

Thank you very much for your generous comments. We appreciate you recognized the novelties in our work. We apologize for all the confusion from mislabeled plots and complex phrases. We have included a point-by-point response. The line numbers mentioned correspond to the revised manuscript with tracked changes. The corrected plots, as well as responses, have been incorporated in the updated version of our manuscript.

General comments: I have two main issues with the paper. The first is that SIF contains a component of PAR while reflectance does not. This makes it a bit unfair to compare any of the reflectance- based quantities directly with GPPmax (similarly affected by PAR as is SIF) and also to derive the PLSR reconstruction of GPPmax without consideration of PAR. It may be fine to do the reconstruction of other quantities (related to pigments) without account of PAR, but generally not GPPmax. A much better result may come from normalizing GPPmax with respect to PAR (or daily averaged PAR or daily averaged potential PAR) and performing the reconstruction on this quantity. This is particularly important in Fig. 9. Some statements may need to be modified after taking into account PAR (e.g., paragraph starting on L. 320).

We apologize that we did not properly introduce the relative SIF in the introduction. In figure 5(d) and figure 9, we accounted for PAR impact on SIF by using relative SIF, which is SIF normalized by the reflected near-infrared. We have rewritten the introduction as well as methods on relative SIF to clarify this (L52-53, L82-90, 152-155).

We agree with you that normalization is necessary. We used Eq. (1) to derive light use efficiency (LUE), which is inevitably conditioned on PAR and fPAR. Benefiting from the APAR measurements we have, we can use in situ APAR as the normalization factor instead of PAR and presumed fPAR in previous studies. According to the light response curve of photosynthesis (figure 2), we summarized three different scenarios LUE: 1) light-limited (low light, $LUE_{lightL}$); 2) carboxylation rate-limited (high light, $GPP_{max}$); and 3) daily average ($LUE_{total}$).

$LUE_{lightL}$ is the fitted slope of GPP and APAR when PAR is between 100-500 $\mu$mol m$_{-2}$ s$_{-1}$. The fit was forced to go through the origin as the equation has no intercept.

$LUE_{total}$ is the daily average of GPP/APAR during the day.

Theoretically, we could calculate $GPP_{max}$ similar to what we did for $LUE_{lightL}$, i.e. regressing GPP against APAR when PAR is saturated. Unfortunately, there is a 26-day gap in APAR measurement in the beginning period of our study. We could also normalize GPP by PAR, as you suggested. Yet, it requires assumptions about fPAR, which brings uncertainties.

Considering GPP often flats out when PAR is greater than 1000 $\mu mol\ m_{-2}\ s_{-1}$, the slope of fits could be biased by the spread of data point. Thus, we defined $GPP_{max}$, the average GPP within a moderately narrow window of PAR (1000-1500 $\mu mol\ m_{-2}\ s_{-1}$), to represent the LUE when the carboxylation rate is limited. In this way, we achieved normalizing GPP without missing more data because of APAR.

Those two different definitions of $GPP_{max}$ are significantly linearly correlated (in fig.AC2.1 in this response document). If $GPP_{max}$ is defined as the average of normalized GPP by PAR (y-axis), the analyses on the seasonal cycle will differ little from $GPP_{max}$ defined as mean GPP at high PAR (x-axis). Although GPP normalized by PAR results in the correct unit of LUE, it is easily mistaken as fPAR has been considered. To avoid this confusion, we chose to use mean GPP at PAR between 1000 and 1500 $\mu mol\ m_{-2}\ s_{-1}$.

[Figure]

Figure.AC2.1 Two ways to calculate $GPP_{max}$.

We also tested the LUE defined in three scenarios: $LUE_{lightL}$, $GPP_{max}$, and $LUE_{total}$ for all the analyses in the manuscript. They behave similarly. We considered that needles spent more time at highlight intensity in the daytime. Hence, we decided to use $GPP_{max}$ as the only proxy of LUE in the main text. We kept the other two matrices to the supplementary. The labels were misleading in the main text. We have cleaned them up in the updated draft (Fig. 3).

The second comment relates to the terminology around the component analysis. It would be helpful if the reflectance can be written in the form of equations explaining the decomposition. Then it may be more clear to express what is being plotted. In my understanding of the terminology, a coefficient should be a number multiplied by a particular spectral component to reconstruct a given spectrum. The term temporal component is confusing to me as this is not what has been decomposed, but rather it is the coefficient of a given spectral component to reconstruct a spectrum observed at a particular time if I have understood correctly. The labeling of Fig. 6 is particularly difficult to understand since panel (a) shows much more than PLSR coefficients. The line labeled GPP_max would be more clear if it had PLSR coefficient in the label.

Thank you for the advice. We have changed "temporal components" to "temporal loading" (Eq. (4) and L201). We added the following expressions to the draft:

ICA:

$$-log(R_{\lambda,\text{DOY}}) = \sum_{i=1,2,3} (\text{spectral component}\,^{i}_{\lambda} \cdot \text{temporal loading}\,^{i}_{\text{DOY}}).$$

PLSR:

$$\text{GPP}_{\text{max,DOY}} = -log(R_{\lambda,\text{DOY}}) \times \text{PLSR coefficient}_{\lambda}^{\text{GPP}_{\text{max}}}.$$

Specific comments:

L. 118. VI's are normalized such that at least some of the solar geometry effects are removed. This may not be the case for reflectance in general.

Thank you for pointing this out. We agree the experiment related to this sentence is not enough to clarify the BRDF impact on reflectance. Thus, we did a PLSR analysis on individual measurements of phase angle and reflectance for 3 summer days (2017-7-1 to 2017-7-3), such as fig.AC2.2.

[Figure]

Figure.AC2.2 PLSR analysis on phase angle and reflectance.

Indeed, the reflectance has different sensitivities to the phase angle. However, the poor correlation of PLSR reconstructed phase angle and the measurement one suggests the variations in phase angle should not be the critical factor for the change in reflectance. In our manuscript, we primarily removed the bi-directional impact by averaging all the individual reflectance that was measured at different solar geometry and viewing geometry. This analysis has been referred in L128-129 in the revision.

L. 142. Normalized at which wavelength(s) exactly? I'm not sure I agree that normalizing by reflected radiation is going to properly "account for the complexity of signal due to canopy structure" at this particular site. Please provide more justification of this statement.

Thanks for pointing out this misleading sentence. SIF was normalized by reflected near-infrared radiation (755 nm) to account for the sunlit/shaded fraction within our observation FOV. Benefiting from the small FOV of PhotoSpec, we think the complexity of canopy structure has minimal impact on our signal. We have clarified the statement in L85-90.

Since it is mentioned on L. 183 that 3 components explain more than 99.99% of variance, can you state how much variance is explained by each of the components shown.

Because ICA minimizes the dependencies of the second-order moment (variance) and higher, the randomness during the minimization makes the explained variance and order of individual component unclear. In our calculation, the ICA algorithm reduced the dimension of the input matrix by eigenvalue decomposition first, from which the first three second-order independent/orthogonal components yielded 99.99% of the variance. Then, the algorithm extracted the independent components of high-order moments from

these orthogonal components. We have included this explanation in section 4 in the supplementary material.

L. 261, it is mentioned that $T_{air}$ and VPD covary with $GPP_{max}$. Please provide some numbers here such as correlations.

We have cross referenced the figures in the supplementary in the draft (L246, L252) while we also included the following the Pearson-r2 value of correlations.

During the growing season:

$GPP_{max}$ and VPD = 0.34

$GPP_{max}$ and $T_{air}$ = 0.24

PRI and VPD = 0.04

PRI and $T_{air}$ = 0.02

I am confused as to what is meant by "short-term" on line 262 as a few lines above it is said to be the smoothest. Some may take short-term to mean daily. In that case, we wouldn't expect it to be smooth.

Thanks for the feedback. We meant to use "short-term" and smoothness to refer day to day or sub-seasonal variations. We have revised the paragraph with a consistent description in the updated manuscript (L305-309).

Have you definitively shown here that the green band captures variations in LUE? Isn't this only inferred?

Yes, we inferred the green band capturing LUE by the good performance of GCC. We changed the word "show" to "suggest" (L294).

Fig. 5, something appears incorrect with the r2 value shown in panel (f).

We found a bug in the calculation and corrected it (Fig. 5). The correct r2 values of $GPP_{max}$ with CCI, PRI, GCC, relative SIF, NDVI, and NIRv are 0.85, 0.81, 0.73, 0.87, 0.06, and 0.40, respectively.

L. 319, What exactly is meant by diurnal? The most common use of this word in my

field pertains to "of or during the day" which is commonly taken to mean sub-daily.

> Yes, it means day-to-day/sub-daily variations here, similar to the previous comment. we have made sure the wording is accurate and consistent (L353-357).

Same paragraph: In this work, it is not definitively shown that SIF tracks seasonal or diurnal variations better than reflectance. While SIF shows slightly higher r2, the differences were not shown to be statistically significant. It is curious that CCI gives a higher r2 value with respect to GPPmax than the PLSR analysis and the PRI gives the same value as PLSR. Also the sample of points looks different in Figs. 5 and 6.

> Thanks for the suggestion. The phrase appears to make readers think we are competing with existing methods, which is misleading. Our study focuses on mechanistically explaining where, when, and why certain wavelength regions are sensitive to canopy LUE seasonality. Thus, a similar performance from CCI and PLSR is expected as CCI uses the most sensitivity band. Although SIF represents a different process from the reflectance-based methods the good performance of relative SIF has been discussed in Magney et al., 2019, which makes Niwot Ridge a very interesting site to study.
>
> After we corrected the calculation, relative SIF and PLSR (r2= 0.87) have the highest r2 compared to other indices, although not significantly. CCI and PRI are comparable but less correlated with respect to $GPP_{max}$ than PLSR. The different reference bands used by CCI and PRI might cause the slight difference in their correlation with PLSR.
>
> The difference in Figs. 5 and 6 were because We only plotted the observed $GPP_{max}$ when the reflectance is also available in Fig 6, while all the observed $GPP_{max}$ was plotted in Fig 5. In the revised draft, these two plots are consistent as Fig 5.

Fig. 9: There is no goodness of fit metric here relative to measurement uncertainties. To make it more clear the fits should be shown with the observations and the residuals and fit properly evaluated with standard metrics. Otherwise the differences are not convincing.

> Thank you for the suggestion. We attached a plot of individual fittings and their evaluations (fig.AC2.3, Fig.D1). The fitted curve has been expressed as the derivation in Appendix D. The pearson-r2 and p values listed in each subplot were calculated from the correlation of observed and fitted variables. The residual was calculated as the average L2 norm of the difference between observed and fitted variables normalized by the observation. The fittings are overall good. Because the ICA component lacks a clear sigmoid shape, ICA has a larger residual.

[Figure]

Figure.AC2.3. Individual sigmoid fits of timeseries of interest. The fitted curves are expressed in the format as appendix D. The pearson-$r_2$ and p values are for the observed and fitted variables. The residual is

$$\frac{1}{n}\sum_i (\frac{x-\hat{x}}{x})^2 ,$$

where x is the observed value and $\hat{x}$ is the fitted value.

L. 334, I may have missed something but I didn't see how this feature was shown to be directly related to LUE in the paper. This may be inferred but it wasn't directly shown.

We are now more precise in phrasing. The sentence (L371-373) has been changed to "The main spectral feature centered around 530 nm is most important for *inferring* the seasonal cycle of reflectance (400 – 900 nm) and LUE, which corresponds to changes in carotenoid content."

Detailed comments:

There are a number of typos that need to be fixed, for example subscripts (L. 1, L. 226, L. 234, Fig. 3, Fig. 6 panel (b)).

Thanks a lot for catching them. We have corrected the typos and made sure all the subscripts are coded correctly in the revision (L1, L239, L244, Fig. 3, and Fig. 6(b)).

There are a number of statements that need to be clarified or corrected for language (see L. 2, for example, "Estimating . . . is a primary uncertainty" would be better phrased as "Estimation of...corresponds to a primary source of uncertainty" or similar). See also lines 11-12 (unclear sentence, L. 14, etc.).

We have rephrased those unclear sentences and be more precise (L2, L10-13).

Line 21: Satellites do not measure GPP, rather GPP can be inferred and usually make use of other data.

The sentence is now more precise in the updated draft (L24).

It's a little confusing in L. 159 to start with "To implement Eq. (1), then define a particular case for Eq. (1). Please rephrase.

We rephrased the sentence as "We followed the format of Eq. (1) to define light-limited LUE ($LUE_{lightL}$) as…" This paragraph has been presented in section 2.2 in the supplementary instead of main text.

L. 164-166. It's not clear at this point what the meteorological data are included for. L. 165 would be more clear to say that daily mean . . . were computed from . . . .

We replaced "meteorological variables" with "Air Temperature ($T_{air}$) and Vapor Pressure Deficit (VPD)" in L148. And, rephrase L164-166 as daily $T_{air}$ and VPD were computed from averaging half-hourly $T_{air}$ and VPD when…" The rephrased sentences are in L176-178 in the revised manuscript.

The first reviewer was unclear about what LUEs/GPPmax is. I think I figured it out but it took a lot of time and was very unclear.

Thanks for taking the time to sort it out. We apologize for the confusion. In the updated draft, we have clarified the usage of acronym LUE with $GPP_{max}$ as the only proxy for LUE in the main text. The results for $LUE_{lightL}$ and $LUE_{total}$ are put in the supplementary.

Line 217. Is LUElight the same as LUElightL defined above?

Yes. Sorry about the typo.

L. 226: Confusion regarding Fig. 3 (not S3) but also most of which is repeated in Fig. S2 but with the lines that are referred to in Fig. 3. Suggest to include only one figure with all the lines (in the main manuscript). A similar thing happens with Fig. 9 and D1. Suggest to include only one of these.

We have only included Fig.3 and Fig.9 in the main draft and eliminated Fig.S2 and Fig.D1.

Fig. 3: Specify that these are daily-averaged quantities?

We have specified them in both the caption in Fig. 3.

Fig. 4 caption: Only the 2nd component shows the carotenoid Jacobian.

Yes, because we want to emphasize the similarity of component 1 with the chlorophyll jacobian only. We have corrected the caption in Fig.4 and made sure it is consistent with the plot.

Sect. 3.2, first par. There is a lot of information in this paragraph. It might be more effective if it was split up.

Yes, we agree with you that is too tedious to read. We have split it into paragraphs and rephrased with the equations of ICA so that it is more clear.

L. 268, the word "thus" here is confusing.

We have removed "thus" from the sentence (L293).

Fig. 6: Panel (a) labeling is very confusing. First, the title of panel (a) as well as the label on right side is confusing as these are not really coefficients are they, they are either components of combinations of components (caption is also unclear)? The caption states that the overlaid solid line is the 2nd ICA component, but there are two solid lines. The blue line in the legend is labeled as GPPmax, but it isn't really GPPmax as labeled in the bottom. Would suggest to just remove the titles of both panels.

Thank you for pointing this out. In fig. 6(a), we have changed the label of "GPP$_{max}$" to "PLSR coefficient". The blue curve indeed is the coefficient when the PLSR is written in a fashion of linear model, such as:

$$\mathrm{GPP_{max,DOY}} = -log(R_{\lambda,\mathrm{DOY}}) \times \mathrm{PLSR\ coefficient}_{\lambda}^{\mathrm{GPP_{max}}}.$$

However, the y-axis on the right is only for PLSR coefficient not for ICA. We clarified that both Jacobians and ICA component were scaled in the caption in Fig. 6.

The solid blue line is the PLSR coefficient. The solid orange line is the 2$_{nd}$ ICA component. We have clarified them in the revised draft. The titles were also removed.

L. 297 should be "support".

> We have corrected it (L333).

It would be better to subscript the small letters in Cchl, Ccar, and Cant.

> We have subscripted them in the revision.

Fig. B1 caption should say theoretical maximum (or clear sky)

> We have changed it to "theoretical maximum" in the revision (L410 and Fig. B1).

[revised manuscript text omitted]